# Host-inherent variability influences the transcriptional response of *Staphylococcus aureus* during *in vivo* infection

Robert Thänert[1], Oliver Goldmann[1], Andreas Beineke[2] & Eva Medina[1]

The rise of antibiotic resistance calls for alternative strategies to treat bacterial infections. One attractive strategy is to directly target bacterial virulence factors with anti-virulence drugs. The expression of virulence traits by pathogens is, however, not constitutive but rather induced by the level of stress encountered within the host. Here we use dual RNA sequencing (RNA-seq) to show that intrinsic variability in the level of host resistance greatly affects the pathogen's transcriptome *in vivo*. Through analysis of the transcriptional profiles of host and pathogen during *Staphylococcus aureus* infection of two mouse strains, shown to be susceptible (A/J) or resistant (C57BL/6) to the pathogen, we demonstrate that the expression of virulence factors is dependent on the encountered host resistance. We furthermore provide evidence that this dependence strongly influences the efficacy of anti-virulence strategies, highlighting a potential limitation for the implementation of these strategies.

[1] Infection Immunology Research Group, Helmholtz Centre for Infection Research, Inhoffenstrasse 7, 38124 Braunschweig, Germany. [2] Institute for Pathology, University of Veterinary Medicine Hannover, Bünteweg 17, 30559 Hannover, Germany. Correspondence and requests for materials should be addressed to E.M. (email: eva.medina@helmholtz-hzi.de).

The rise of antimicrobial resistance is one of the most challenging problems in modern medicine, causing an increase in morbidity and mortality associated with common bacterial infections[1]. While available antibiotics are loosing their effectiveness, the introduction of novel bactericidal or bacteriostatic antibiotics cannot be considered a long-term solution because it is eventually followed by the emergence of resistant bacterial clones that become increasingly prevalent under selective drug pressure. Consequently, there is a pressing need for new anti-infective agents that do not impose similar levels of selection pressure on pathogens as classical antibiotics. In this regard, alternative approaches based on attenuating bacterial pathogenesis by targeting bacterial virulence, the so-called 'anti-virulence' strategies, are emerging as promising tools for the treatment of infections[2].

Bacterial pathogens express a large repertoire of different virulence factors to survive under the adverse conditions imposed by the host environment. Thus, anti-virulence strategies have been proposed that specifically target bacterial toxins produced by the pathogen to evade host defenses[3], bacterial factors mediating adhesion to the host[4], secretion systems[5] as well as regulatory systems[6] and quorum-sensing signalling[7]. The key feature of anti-virulence drugs is the attenuation of the pathogen's virulence to aid clearance by the host's immune defenses[2]. These drugs seem attractive, because it is believed that not killing the pathogen directly exerts less selective pressure for the development of resistance[2]. However, such an approach will only confer therapeutic benefit if the targeted virulence factor(s) are actually expressed by the bacterium during infection and if the natural defense mechanisms of the host are strong enough to clear the pathogen, weakened by the anti-virulence treatment.

Bacterial pathogenesis, on the other hand, is strongly influenced by the strength of the host immune defense. For example, avirulent microorganisms can be pathogenic for immunocompromised hosts, whereas virulent microorganisms can be nonpathogenic in immune hosts[8]. This situation is further complicated by the fact that, in addition to the immune status, inherent characteristics of the host, such as the genetic background, significantly influence the capability of the immune system to overcome invading pathogens. Thus, the response to a specific pathogen can range from weak in susceptible hosts, causing severe infections, to strong in more resistant individuals, resulting in milder diseases. These differences imply that pathogens will encounter stronger immune pressure in resistant than in susceptible hosts and virulence factors that are essential for counteracting a weak immune response may not be the same as those required under stronger immune pressure in resistant hosts. Therefore, the dependence of virulence factor expression on host resistance is a potential limitation for the effectiveness of anti-virulence drugs.

Here we investigate how intrinsic variability of host resistance to a pathogen affects the expression of virulence factors needed to successfully infect the host. We use Staphylococcus aureus, a human pathogen that can cause severe invasive infections[9] and is notorious for its capacity to develop antibiotic resistances. These characteristics make S. aureus one of the most dangerous and intractable infectious pathogens worldwide[10]. Despite numerous attempts to develop a vaccine that can prevent S. aureus infections, none of the vaccine candidates tested in clinical trials has succeeded so far[11]. This failure, in combination with the increase of antibiotic resistance, has lead to an intensification of efforts to search for alternative treatment approaches in recent years. In this regards, anti-virulence strategies targeting crucial pathogenicity factors produced by S. aureus during infection have been proposed as an attractive therapeutic option[12,13]. However, since the outcome of S. aureus infection is strongly influenced by

the host factors such as racial origin, age and genetic makeup[14,15], the search for anti-virulence targets in S. aureus has to consider the inherent variability of the host responses to infection. Similar to humans, variability in the host response to S. aureus has been also observed among different inbred strains of mice[16]. While some mouse strains (for example, A/J and DBA/2) are very susceptible to S. aureus infection, C57BL/6 mice are highly resistant and survive a bacterial dose that rapidly kills mice from susceptible strains[16]. These differences in the capacity to control S. aureus infection provide a unique experimental system to explore the extent to which intrinsic host variability affects the expression of bacterial virulence factors during infection.

In this study, we use dual RNA sequencing (RNA-seq), and two mouse strains previously shown to display differential susceptibility to S. aureus infection (strain A/J is susceptible and strain C57BL/6 is resistant)[16,17], to investigate how the intrinsic variability of host resistance to a pathogen affects the expression of virulence factors. Dual RNA-seq enables the simultaneous determination of the transcriptional response of the host and pathogen during infection and does not require physical separation of prokaryotic and eukaryotic RNA since the sequencing reads can be assigned to the host or to the pathogen genomes by in silico analysis[18]. Although dual RNA-seq has been successfully applied to characterize the transcriptional signature of bacteria and host in several in vitro infection systems[14,15,19,20], the study presented here is one of the first using this technology in an in vivo system. Our results demonstrate the impact of the host genetic background on the transcriptional response of S. aureus during infection, and provide experimental evidence that host-dependent bacterial expression of virulence factors is a potential limitation for the efficacy of anti-virulence therapies.

## Results

**C57BL/6 and A/J mice differ in their resistance to S. aureus.** For this study, we selected two strains of mice that differ in their susceptibility to S. aureus infection[16,17]. Whereas A/J mice are very vulnerable to S. aureus, resulting in a significantly increased bacterial growth in the kidneys (Fig. 1a) and liver (Fig. 1b) and fatal infection outcome (Fig. 1c), C57BL/6 mice exhibit greater resistance to S. aureus and were capable to significantly restrict bacterial growth in the kidneys (Fig. 1a) and liver (Fig. 1b) with all mice surviving (Fig. 1c). Therefore, these two mouse strains provide an excellent tool for investigating the extent to which variability in the host response to infection affects the transcriptional response of S. aureus within the host.

**Dual RNA-seq analysis of S. aureus and infected host tissue.** A dual RNA-seq approach that enables simultaneous transcriptional profiling of bacteria and host tissue was used to characterize the host response to infection and investigate the impact of different levels of host resistance on the transcriptional response of S. aureus. The schematic representation of the experimental design is shown in Fig. 1d. Total RNA including host and pathogen RNA, was isolated from the kidneys of A/J and C57BL/6 mice at 48 h of infection and analysed by Illumina deep sequencing. Between 9 and 21 millions reads were uniquely aligned to the reference genome of Mus musculus assembly GRCm38.p3 (GCA_000001635.5), while between 32,228 and 5 millions reads were uniquely mapped to the revised reference genome of S. aureus strain 8325-4 (ref. 21).

**Transcriptome analysis of S. aureus-infected mice.** Hierarchical clustering (Supplementary Fig. 1a) and principal component analysis (PCA) (Supplementary Fig. 1b) of gene expression datasets from infected A/J and C57BL/6 mice, as well as of

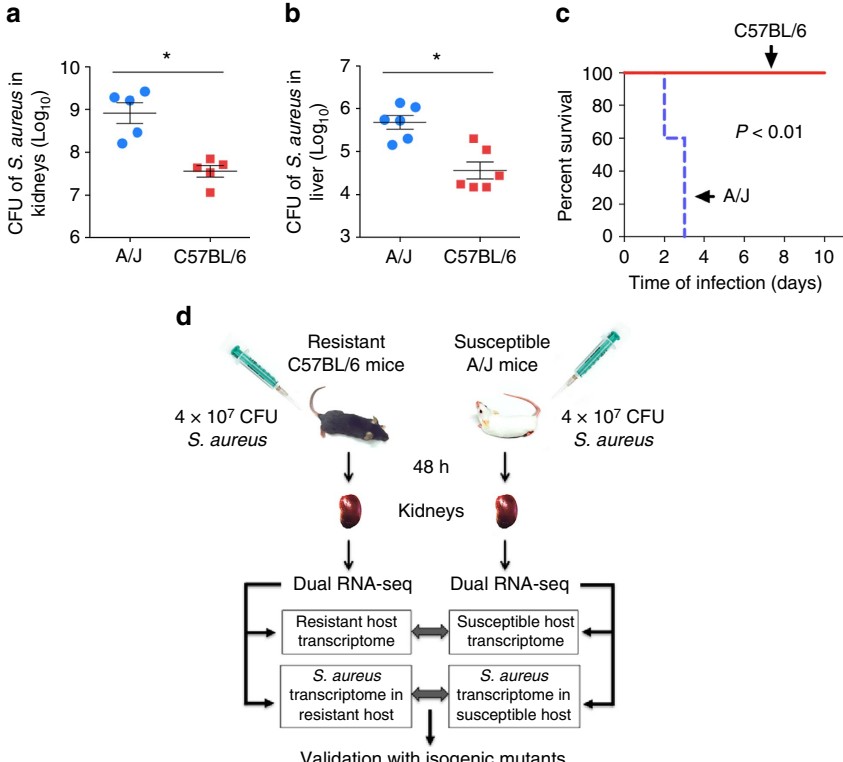

**Figure 1 | A/J and C57BL/6 mice exhibit opposed levels of resistance to *S. aureus*.** Bacterial loads in the kidneys (**a**) and liver (**b**) of A/J and C57BL/6 mice at 48 h after intravenous inoculation with $4 \times 10^7$ CFU of *S. aureus* SH1000. Each symbol represents the bacterial counts determined in an individual mouse and the horizontal lines represent the average ± s.d. for each mouse strain. One representative experiment out of three independent experiments is shown ($n = 6$, *t*-test, *$P < 0.05$). (**c**) Survival curves of A/J and C57BL/6 mice intravenously infected with $4 \times 10^7$ CFU of *S. aureus* SH1000 ($n = 5$, log-rank test, $P < 0.01$). (**d**) Schematic summary of the experimental design for dual RNA-seq analysis. Susceptible A/J mice and resistant C57BL/6 mice were infected intravenously with $4 \times 10^7$ CFU of *S. aureus* SH1000, their kidneys removed at 48 h after bacterial inoculation and subjected to dual RNA-seq analysis to simultaneously determine the gene expression profile of the host and pathogen in the same sample. The genes differentially expressed by *S. aureus* in A/J and C57BL/6 mice were identified and related to the infection-associated transcriptional response of the corresponding mouse strain. The effect of targeting a virulence factor differentially expressed by *S. aureus* SH1000 between infection of A/J and C57BL/6 mice was also determined.

tissue from uninfected control mice, demonstrated high within-group reproducibility and substantial between-group differences. Pairwise comparison by permutational multivariate analysis of variance (PERMANOVA) revealed that the differences between all groups were statistically significant ($P < 0.05$, Supplementary Table 1). Differential gene expression analysis using DESeq2 (ref. 22) identified a total of 5,540 differentially expressed genes (DEGs) between uninfected and *S. aureus*-infected A/J mice, of which 2,756 were found to be up-regulated in response to infection, while 2,784 were downregulated (Supplementary Data 1). In C57BL/6 mice, 3,559 genes were differentially expressed between uninfected and *S. aureus*-infected mice, of which 1,758 were upregulated and 1,801 were down-regulated in response to infection (Supplementary Data 2). The numbers of unique and overlapping DEGs between *S. aureus*-infected A/J and C57BL/6 mice are shown in Fig. 2a.

Functional classification of the DEGs using KEGG (Kyoto Encyclopedia of Genes and Genomes) pathway enrichment analysis, revealed that a large group of host genes with increased expression in response to infection in both A/J and C57BL/6 mice belonged to the groups 'cytokine-cytokine receptor interaction' and 'chemokine signalling pathway' (Fig. 2b). Particularly, genes encoding inflammatory cytokines such as IL-6, IL-1α, IL-1β and TNF-α as well as chemokines involved in the chemoattraction of monocytes/macrophages such as Cxcl1, Cxcl2 and Cxcl3, were upregulated in both A/J and C57BL/6 mice in response to *S. aureus* infection (Supplementary Data 1 and 2). Also, host

genes encoding acute phase proteins such as Saa1, Saa2, haptoglobin and the calcium-binding proteins S100a8 and S100a9 were highly induced in infected A/J and C57BL/6 mice. Although the global analysis of the transcriptional data suggested that a 'core' set of inflammation-related genes was highly expressed in both A/J and C57BL/6 mice in response to *S. aureus* infection, the average fold change of expression in this set of genes was markedly higher in infected A/J than in infected C57BL/6 mice (Fig. 2). This suggested that A/J mice developed a more intense systemic inflammation than C57BL/6 mice in response to *S. aureus* infection, which is indicative of severe sepsis leading to death. Besides the systemic hyperinflammation, the increased expression of the gene encoding the coagulation activator tissue factor (*F3*) and of the gene encoding the fibrinolysis inhibitor PAI-1 (*Serpine1*) in A/J mice (Supplementary Data 1), revealed a net pro-coagulant status that is typical for severe sepsis[23]. Altered coagulation, coupled with microvascular dysfunction occurring during sepsis, decreases tissue perfusion. This leads to perturbations of oxygen supply, resulting in tissue hypoxia and the activation of the hypoxia-inducible factor alpha encoded by *Hif1a*[24,25]. The significant induction of *Hif1a* observed in A/J, but not in C57BL/6 mice, in response to infection (Supplementary Tables 2 and 3), indicated more severe tissue hypoxia in the kidneys of infected A/J mice than in those of infected C57BL/6 mice. Furthermore, the genes encoding arginase 1 (*Arg1*) and arginase 2 (*Arg2*) were expressed to a greater extent in infected A/J than in infected C57BL/6 mice

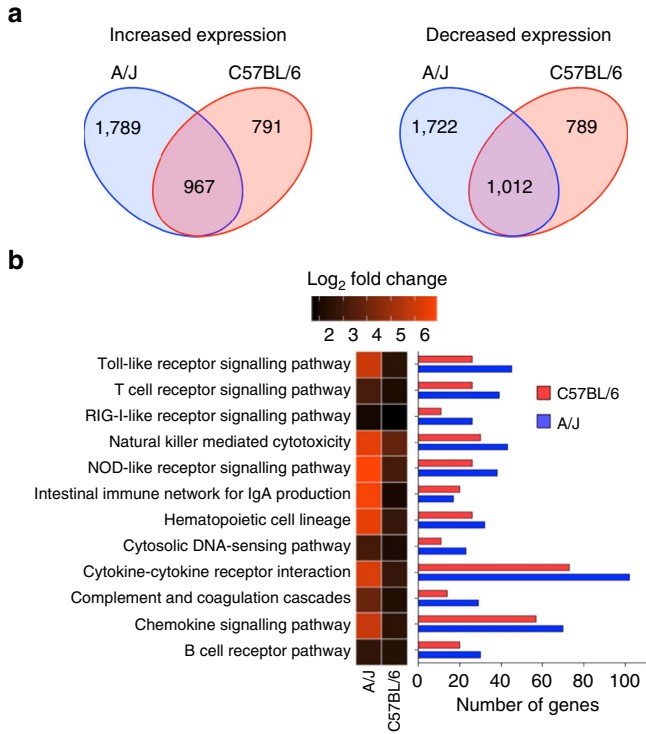

**Figure 2 | Gene expression analysis of *S. aureus*-infected kidneys from A/J and C57BL/6 mice.** (**a**) Venn diagram showing the number of DEGs with increased (left) or decreased (right) expression in response to *S. aureus* infection that are unique or common between A/J and C57BL/6 mice. (**b**) Heat map of the mean log$_2$-fold change of gene expression of the significantly up-regulated genes determined between *S. aureus*-infected versus uninfected A/J and C57BL/6 mice within the KEGG 'immune system' category (left part). The corresponding numbers of significantly up-regulated genes in response to *S. aureus* infection in A/J (blue bars) and C57BL/6 (red bars) are shown in the right part of the figure.

(Supplementary Data 1). Arginase can contribute to endothelial cell dysfunction by depleting extra-cellular L-arginine and nitric oxide (NO) bioavailability, leading to endothelial nitric oxide synthase uncoupling and consequently to the production of high levels of harmful reactive oxygen species (ROS)[26]. Taken together, these findings suggest that the micro-environment in the infected tissue is highly different between A/J and C57BL/6 mice, which could significantly affect the expression of virulence determinants by *S. aureus*.

**S. aureus transcriptome during infection of A/J or C57BL/6.** In parallel, we analysed the transcriptome of *S. aureus* during the infection of resistant C57BL/6 and susceptible A/J mice to determine the impact of the different physiological conditions present at the site of infection on the pathogen's transcriptional response. Hierarchical ordination (Supplementary Fig. 2a) and PCA (Supplementary Fig. 2b) showed high within-group reproducibility, while PERMANOVA demonstrated that the transcriptional response of *S. aureus* infecting susceptible A/J mice differed significantly from that of *S. aureus* infecting resistant C57BL/6 mice (Supplementary Table 2). Transcript abundance was determined by normalizing the number of raw reads in each data set for gene length and expressed as transcripts per Kilobases per Million (TPM) (Supplementary Data 3). A total of 85 genes were identified as differentially expressed (probability value $\geq 0.95$) by *S. aureus* between infection of A/J and infection of C57BL/6 mice using NOISeq (Supplementary Data 4). Of those, transcripts of 20 genes were more abundant in *S. aureus* infecting A/J mice (Fig. 3a, Table 1, Supplementary Table 4), 65 genes exhibited greater expression in *S. aureus* during infection of C57BL/6 mice (Fig. 3a, Table 2, Supplementary Table 5) and 594 were expressed at a similar level by *S. aureus* in A/J and C57BL/6 mice (Fig. 3a, Supplementary Data 4).

One of the most prominent operons expressed by *S. aureus* to a greater extent during infection of A/J mice, than during infection of C57BL/6 mice, was the arc operon, which encodes the arginine

**Table 1 | DEGs between *S. aureus* infecting A/J and C57BL/6 mice with greater transcript abundance during infection of A/J mice.**

| Locus tag | Gene symbol | Description |
|---|---|---|
| SAOUHSC_00845 | | Hypothetical |
| SAOUHSC_02853 | | Hypothetical |
| SAOUHSC_00371 | *yflT* | Hypothetical |
| SAOUHSC_02964 | *arcR* | Hypothetical |
| SAOUHSC_01477 | | Hypothetical |
| SAOUHSC_01969 | *gvpP* | Hypothetical |
| SAOUHSC_00101 | *drm* | Phosphopentomutase |
| SAOUHSC_01181 | *xynA* | Hypothetical |
| SAOUHSC_02967 | *arcD* | Arginine/ornithine antiporter |
| SAOUHSC_01191 | *rpmB* | 50S ribosomal protein L28 |
| SAOUHSC_00686 | | Hypothetical |
| SAOUHSC_01803 | *aapA* | Hypothetical |
| SAOUHSC_02862 | *clpL* | ATP-dependent Clp protease, ATP-binding subunit ClpC |
| SAOUHSC_01403 | *cspA* | Cold shock protein |
| SAOUHSC_02850 | *cidB* | Hypothetical |
| SAOUHSC_01002 | *qoxB* | quinol oxidase AA3 subunit II |
| SAOUHSC_01024 | *graF* | Hypothetical |
| SAOUHSC_02702 | | Hypothetical |
| SAOUHSC_02697 | *tcyC* | Amino acid ABC transporter ATP-binding protein |
| SAOUHSC_02665 | | Hypothetical |

DEG, differentially expressed gene.
The complete data of the differentially expressed genes with higher expression by *S. aureus* during infection of the susceptible A/J mice are displayed in Supplementary Table 4.

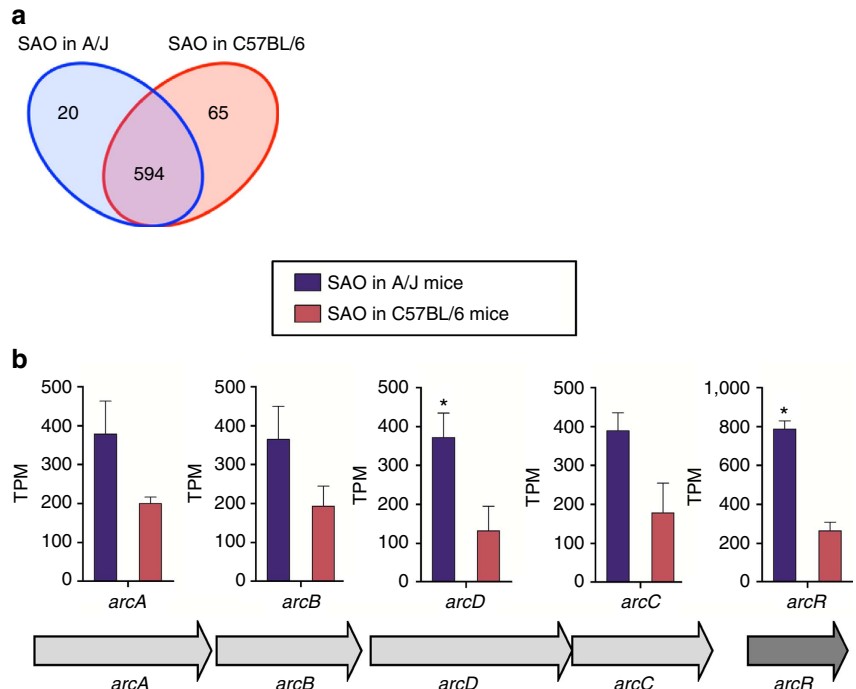

**Figure 3 | Gene expression analysis of *S. aureus* during infection of A/J and C57BL/6 mice.** (**a**) Venn diagram showing the number of unique and common expressed genes between *S. aureus* infecting A/J and C57BL/6 mice based on differential gene expression analysis determined with NOISeq. (**b**) Gene composition and organization of the genes of the ADI operon and their level of expression in *S. aureus* during infection of A/J (blue bars) or C57BL/6 (red bars) mice. Each bar represents the mean of TPM ± s.d. of triplicates.

deiminase (ADI) system (Fig. 3b, Supplementary Data 4). The arc operon comprises the genes coding for arginine deiminase (*arcA*), ornithine transcarbamylase (*arcB*), carbamate kinase (*arcC*), the arginine/ornithine antiporter (*arcD*) and the transcriptional regulator ArcR (*arcR*). These enzymes catalyse the conversion of arginine to ornithine, ammonia, and $CO_2$, while producing ATP. This not only enables *S. aureus* to utilize arginine as an energy source under anaerobic conditions[27] but also generates ammonia, which confers protection against acidic stress[28]. The expression of all genes of the ADI pathway was greater in *S. aureus* during infection of A/J than during infection of C57BL/6 mice, even though only the differences in the level of *arcR* and *arcD* expression reached statistical significance (false discovery rate (FDR) < 0.05, NOISeq analysis), (Fig. 3b, Supplementary Data 4).

Other genes, which were differentially expressed by *S. aureus* during infection of A/J and C57BL/6 mice, were those encoding proteins for amino acid transport and biosynthesis. Thus, transcripts of the genes encoding a hypothetical cysteine permease (*tcyC*), a hypothetical lysine permease (*lysP2*), glutamate synthase (*gltB, gltD*) and homoserine dehydrogenase (*dhoM*) were more abundant in *S. aureus* during infection of C57BL/6 mice than during infection of A/J mice (Table 2). Because *de novo* amino acid synthesis pathways and amino acids transport systems are under control of catabolite repression in the presence of preferred carbon sources[29] in *S. aureus*, the increased expression of the genes encoding these pathways during infection of C57BL/6 mice is consistent with more severe nutrient limitation in the tissue microenvironment of C57BL/6 than in the tissue of A/J mice. This nutrient limitation leads to de-repression of genes that enable *S. aureus* to use secondary carbon sources such as amino acids. Furthermore, genes encoding exoproteases such as staphopain (*sspB2*), serine proteases (*sspA, splA, splE* and *splF*), staphostatin B (*sspC*), aureolysin (*aur*) and a cysteine protease

(*sspB*) were also expressed by *S. aureus* to a greater extent in C57BL/6 than in A/J mice (Fig. 4a). These proteases can generate peptides in the extracellular environment that can be imported by the bacterium via specialized transport systems and used to retrieve amino acids[30]. Besides their role in metabolism, extracellular proteases are potent virulence factors that help *S. aureus* to evade the host immune defenses[31,32]. Other genes encoding important virulence factors such as the immunodominant staphylococcal antigen B (*isaB*), the extracellular fibrinogen binding protein Efb (*efb*), which is involved in inhibition of phagocytosis[33], the pore-forming cytolysin alpha-toxin (*hla*), and the amphipathic α-helical phenol-soluble modulins *psma1, psma2* and *psma3*, which can kill host cells by damaging the plasma membrane[34], were also expressed by *S. aureus* to a higher extent during infection of C57BL/6 mice (Fig. 4b). These findings indicate that *S. aureus* express greater levels of virulence factors during infection of C57BL/6 than during infection of A/J mice, which is most probably driven by the different growth phase of the bacteria in the two mouse strains.

The expression of these virulence factors is controlled by regulatory elements such as two-component regulatory systems (TCRS) and transcriptional regulatory systems in response to environmental cues encountered by the bacterium during infection[35]. The staphylococcal quorum-sensing system accessory gene regulator (*agr*) was found highly expressed by *S. aureus* during infection of both A/J and C57BL/6 mice. The *agr* system comprises two divergent transcripts, RNAII and RNAIII, which are under the control of two distinct promoters, P2 and P3, respectively[36]. RNA II encodes the quorum-sensing elements AgrB, AgrD, AgrC and AgrA that represent an autocatalytic sensory transduction system. RNAIII encodes delta-hemolysin (*hld*)[37] and is a major regulator of virulence factors in *S. aureus*, inducing the transcription of various extracellular proteases and toxins[38]. Although the genes encoding the *agr* P2 operon

**Table 2 | DEGs between *S. aureus* infecting A/J and C57BL/6 mice with greater transcript abundance during infection of C57BL/6 mice**

| Locus tag | Gene symbol | Description |
|---|---|---|
| SAOUHSC_02260 | hld | Delta-hemolysin |
| SAOUHSC_00411.1 | psma1 | Alpha phenol soluble modulin |
| SAOUHSC_02566 | sarR | Hypothetical |
| SAOUHSC_02971 | aur | Zinc metalloproteinase aureolysin |
| SAOUHSC_00435 | gltB | Glutamate synthase large subunit |
| SAOUHSC_01788 | thrS | Threonyl-tRNA synthetase |
| SAOUHSC_00987 | sspB | Cysteine protease |
| SAOUHSC_00248 | lytM | Peptidoglycan hydrolase |
| SAOUHSC_02571 | ssaA | Secretory antigen |
| SAOUHSC_00427 | sle1 | Autolysin |
| SAOUHSC_02941 | nrdG | Hypothetical |
| SAOUHSC_01001 | qoxA | Quinol oxidase subunit I |
| SAOUHSC_00964 | | Hypothetical |
| SAOUHSC_00401 | | Hypothetical |
| SAOUHSC_00717 | saeP | Hypothetical |
| SAOUHSC_00741 | nrdI | Ribonucleotide reductase stimulatory protein |
| SAOUHSC_01942 | splA | Serine protease SplA |
| SAOUHSC_00083 | sbnI | Hypothetical |
| SAOUHSC_00348 | rpsF | 30S ribosomal protein S6 |
| SAOUHSC_00436 | gltD | Glutamate synthase subunit beta |
| SAOUHSC_00051 | plc | 1-phosphatidylinositol phosphodiesterase |
| SAOUHSC_00411.2 | psma2 | Alpha phenol soluble modulin |
| SAOUHSC_01121 | hla | Alpha-hemolysin |
| SAOUHSC_00272 | | Hypothetical |
| SAOUHSC_00801 | secG | Preprotein translocase subunit SecG |
| SAOUHSC_01935 | splF | Serine protease SplF |
| SAOUHSC_02369 | rpoE | DNA-directed RNA polymerase subunit delta |
| SAOUHSC_00268 | | Hypothetical |
| SAOUHSC_01110 | efb | Fibrinogen-binding protein-like protein |
| SAOUHSC_01688 | lepA | GTP-binding protein LepA |
| SAOUHSC_02855 | amiD2 | LysM domain-containing protein |
| SAOUHSC_02762 | | Hypothetical |
| SAOUHSC_02114 | dagK | Putative lipid kinase |
| SAOUHSC_02372 | | Hypothetical |
| SAOUHSC_02430 | htsA | ABC transporter periplasmic binding protein |
| SAOUHSC_01320 | dhoM | Homoserine dehydrogenase |
| SAOUHSC_00986 | sspC | Cysteine protease |
| SAOUHSC_00411.3 | psma3 | Alpha phenol soluble modulin |
| SAOUHSC_02112 | | Hypothetical |
| SAOUHSC_02972 | isaB | Immunodominant antigen B |
| SAOUHSC_02885 | | Hypothetical |
| SAOUHSC_01326 | lysP2 | Hypothetical |
| SAOUHSC_02127 | sspB2 | Staphopain thiol proteinase |
| SAOUHSC_01936 | splE | Serine protease SplE |
| SAOUHSC_00728 | ltaS | Hypothetical |
| SAOUHSC_00625 | mnhA | Putative monovalent cation/H+ antiporter subunit A |
| SAOUHSC_02763 | opp-1F | Peptide ABC transporter ATP-binding protein |
| SAOUHSC_00988 | sspA | Glutamyl endopeptidase |
| SAOUHSC_00711 | | Hypothetical |
| SAOUHSC_00561 | vraX | Hypothetical |
| SAOUHSC_02550 | FdhD | Formate dehydrogenase accessory protein |
| SAOUHSC_00875 | ndh2 | Hypothetical |
| SAOUHSC_01359 | mprF | Hypothetical |
| SAOUHSC_01192 | vfrA | Hypothetical |
| SAOUHSC_02887 | isaA | Immunodominant antigen A |
| SAOUHSC_02254 | groEL | chaperonin GroEL |
| SAOUHSC_02485 | rpoA | DNA-directed RNA polymerase subunit alpha |
| SAOUHSC_01462 | gpsB | Hypothetical |
| SAOUHSC_00367 | tcyP | Hypothetical |
| SAOUHSC_01062 | | Hypothetical |
| SAOUHSC_00893 | namA | FMN oxidoreductase |
| SAOUHSC_00144 | ausA | Hypothetical |
| SAOUHSC_00652 | fhuA | Iron compound ABC transporter ATP-binding protein |
| SAOUHSC_02883 | ssaA | LysM domain-containing protein |
| SAOUHSC_01431 | msrB | Methionine sulfoxide reductase B |

DEG, differentially expressed gene.
The complete data of the differentially expressed genes with higher expression by *S. aureus* during infection of the resistant C57BL/6 mice are displayed in Supplementary Table 5.

(agrBDCA) were expressed by *S. aureus* to a similar level in both mouse strains, the expression level of RNAIII/hld was greater during infection of C57BL/6 mice (Fig. 4c, Supplementary Data 4). The gene encoding the transcriptional regulator SarR was also upregulated by *S. aureus* infecting C57BL/6 mice (Fig. 4c, Supplementary Data 4). This could explain the higher level of transcripts encoding proteases and toxins detected in

*S. aureus* during infection of C57BL/6 mice, since both RNAIII (ref. 38) and SarR[39] activate their transcription.

Antimicrobial peptides (AMPs) are an important part of the host innate immune defense against *S. aureus* by directly impairing the integrity of the bacterial cell wall[40]. The bacterial gene encoding phosphatidylglycerol lysyltransferase (mprF), which is part of the cell wall stress stimulon and mediates

resistance to cationic AMPs, by reducing the negative charge of the membrane surface[41], was expressed by *S. aureus* to a larger extent during infection of resistant C57BL/6 than during infection of A/J mice. Similarly, other genes of the cell wall stress stimulon including *vraX*, autolysin encoding genes (*ssaA, lytM, amiD2* and *isaA*), genes involved in cell separation (*gpsB* and *sle1*), *groEL* encoding a chaperonin, and the lipoteichoic acid synthase encoding gene *ltaS* exhibited greater expression levels in *S. aureus* during infection of C57BL/6 mice than during infection of A/J mice. The higher induction of these genes might reflect an adaptive response of *S. aureus* to keep cell wall integrity when confronted with high levels of cell wall damaging agents in the tissue of C57BL/6 mice.

Notably, genes related to central metabolism, iron acquisition, general and oxidative stress responses were expressed by *S. aureus* to a similar extent during infection of A/J and C57BL/6 mice (Supplementary Data 4), indicating that their expression was not influenced by the levels of host resistance to infection.

The RNA-seq results were validated in a selected set of genes from host and pathogen by quantitative reverse transcription–PCR (RT–PCR; Supplementary Fig. 3) and at the protein level by ELISA (Supplementary Fig. 4).

**Effect of decreased host resistance on *S. aureus* transcription.** Next, we determined if depressing specific mechanisms of host resistance in C57BL/6 mice affected the transcriptional response of *S. aureus* during infection. First, we hypothesized that apolipoprotein B (ApoB), the major structural protein of very low-density (VLDL) and low-density lipoproteins (LDL), contributes to the resistance of C57BL/6 mice to *S. aureus* in our infection model, because the gene encoding Apo B (*Apob*) was expressed to a greater extent in the kidneys of C57BL/6 than in the kidneys of A/J mice at both gene expression (Fig. 5a) and protein level (Fig. 5b), before and after infection (Fig. 5c). Furthermore, ApoB has been shown to contribute to the host defense against *S. aureus* in experimental models of skin[42] and respiratory[43] infection by antagonizing the *agr* quorum sensing system of *S. aureus*. To validate this hypothesis, A/J and C57BL/6 mice were treated with 4-aminopyrazolopyrimidine (4-APP), a drug that impairs low-density lipoprotein secretion[44], and subsequently infected intravenously with *S. aureus*. Whereas inhibition of ApoB rendered C57BL/6 mice more susceptible to *S. aureus*, demonstrated by the significantly higher bacterial loads in kidneys at 48 h infection (*P*-value < 0.05, *t*-test), (Fig. 5d), it did not affect the level of susceptibility of A/J mice (Fig. 5d). Next, we investigated if the reduced resistance in C57BL/6 mice after attenuation of ApoB secretion impacted the transcriptional response of *S. aureus* during infection. For this purpose, we compared the expression of a set of virulence-related genes in *S. aureus* infecting 4-APP-treated C57BL/6 mice with the expression of the same set of genes in *S. aureus* infecting C57BL/6 mice treated with vehicle alone. The results show that the level of expression of all genes tested *hld*/RNAIII, *sarR*, *hla*, *sspA*, *aur*, *vraX* and *gltB* was decreased by 4-APP-treatment, while the expression of the hypothetical alanine permease (*aapA*) was higher in *S. aureus* infecting 4-APP-treated than in vehicle-treated C57BL/6 mice (Table 3). These genes were also found to be differentially expressed between *S. aureus* infecting C57BL/6 mice and *S. aureus* infecting C57BL/6 mice deficient in the expression of MyD88, an adaptor molecule that is essential for the signalling of IL-1R/TLR family (Table 3). Since MyD88-deficient mice are more susceptible to *S. aureus* than wild type C57BL/6 mice[45], these observations further demonstrated the remarkable influence of the level of host resistance on the transcriptional response of *S. aureus* during infection.

**Target expression affects efficacy of anti-virulence approaches.** After having demonstrated the influence of the levels of host resistance on both the quality and quantity of *S. aureus* transcriptional response, we sought to determine the consequences of this dependence on the effectiveness of anti-virulence strategies. For this purpose, we assessed the effect of neutralizing a virulence factor that differed in expression between *S. aureus* infecting A/J and C57BL/6 mice on the bacteria fitness during infection. We chose aureolysin, which has been shown to be important for full virulence of *S. aureus* in experimental infection models[46,47] and was expressed to a significantly greater extent by *S. aureus* during infection of C57BL/6 than during infection of A/J mice in our study (FDR < 0.05, NOISeq analysis). C57BL/6 and A/J mice were simultaneously challenged with wild type and an aureolysin-deficient strain of *S. aureus* (wild type, *Δaur*) and the amount of each bacterial strain was determined in the kidneys of infected mice at 48 h of infection. The overall amount of *S. aureus* bacteria counting both wild type and *Δaur* strains was significantly greater ($P = 0.0138$, *t*-test) in the kidneys of A/J ($1.3 \times 10^8 \pm 4.3 \times 10^7$) than in the kidneys of C57BL/6 mice ($1.03 \times 10^7 \pm 5.4 \times 10^6$). Moreover, while a lower amount of *Δaur* than wild type *S. aureus* was recovered from C57BL/6 mice, the amount of *Δaur* recovered from A/J mice was comparable to that of wild type strain (Fig. 6a). Thus, the *Δaur* had a competitive disadvantage when co-administered with wild type *S. aureus* in C57BL/6 mice (mean competitive index for Δaur/wild type *S. aureus* of 0.013), while *Δaur* and wild type *S. aureus* were equally competitive after co-administration in A/J mice (mean competitive index for Δaur/wild type *S. aureus* of 1.595) (Fig. 6b). Taken together, these results indicate that the efficacy of targeting a virulence factor by anti-virulence strategies will strongly depend on its level of expression by the pathogen during infection, which in turn is highly influenced by the intrinsic levels of host resistance to infection.

**Discussion**
Anti-virulence strategies based on attenuation of bacterial pathogenesis by the specific inhibition of virulence factors essential for the pathogen's survival during infection[48], have received increasing attention as novel treatment options for infections caused by antibiotic-resistant pathogens[48]. The concept of anti-virulence therapy is still very much in its infancy and therefore more research is needed to explore its practicability. One important aspect that should be considered carefully when designing anti-virulence strategies is that the expression of virulence traits by the pathogens is not constitutive but rather influenced by the specific environment encountered during infection. Consequently, absent expression of the targeted virulence factors could render anti-virulence strategies completely ineffective. Therefore, it is essential to understand the impact of the wide-ranging, inter-individual variation of the host response on the pathogen's expression of virulence determinants during infection.

Our study supports the idea that the host genetic background affects the transcriptional response of *S. aureus* during infection. The limited capability of the immune defense mechanisms of susceptible A/J mice to control *S. aureus* growth led to the development of an intense inflammatory response, apparent by the disproportionate expression of inflammatory cytokines and damage-associated molecular patterns. Transcriptional reprogramming of the eukaryotic cells, possibly resulting from the concomitant accumulation of acidic products and the lowered oxygen tension (hypoxia) in the infected tissue, involved the induction of *Hif1a* that encodes the central mediator of transcriptional responses to hypoxia HIF-1α[49,50]. Therefore, the

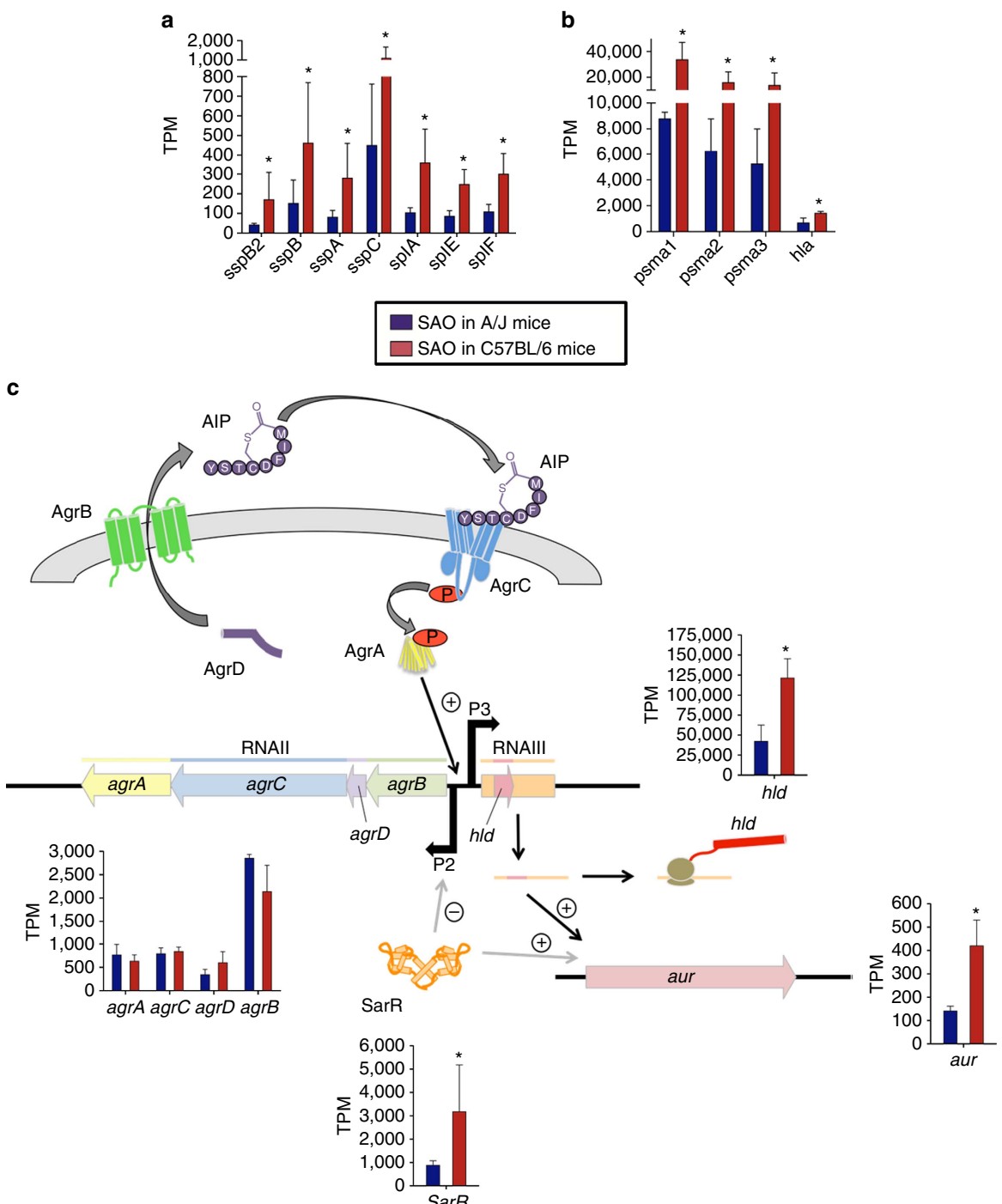

**Figure 4 | Expression of selected genes by *S. aureus* during infection of C57BL/6 or A/J mice.** (**a**) Expression levels of genes encoding proteases. (**b**) Expression levels of genes encoding toxins. (**c**) Schematic representation and level of expression of genes encoding the global regulator Agr and SarR. Red bars show expression values of the genes expressed by *S. aureus* in C57BL/6 mice and blue bars show expression values of the genes expressed by *S. aureus* in A/J mice. Each bar represents the mean TPM ± s.d. of triplicates.

major challenge faced by *S. aureus* in susceptible A/J mice seems to be the adaptation to the adverse conditions imposed by the hyperinflammatory response and hypoxic microenvironments. To survive in the septic A/J mice, *S. aureus* increased expression of the ADI operon (*arcABDCR*), which is generally induced under anaerobic conditions[27] and is important for energy generation, but also protects *S. aureus* against acidic stress[28].

The superior resistance mechanisms of C57BL/6 mice against *S. aureus* enabled a better control of bacterial multiplication.

Therefore, *S. aureus* faced the main challenge of counteracting the powerful host defense mechanisms of the resistant C57BL/6 mice. The transcription data indicated that *S. aureus* responded to the adverse environment encountered within C57BL/6 mice by increasing the expression of cytotoxins and extracellular proteases. Cytotoxins such as alpha-hemolysin (*hla*) and PSMs (*psma1-3* and *hld*) help *S. aureus* to avoid phagocytic killing by inducing pores in the membrane of host cells, leading to cell death[51] and can promote bacterial spreading by disrupting the

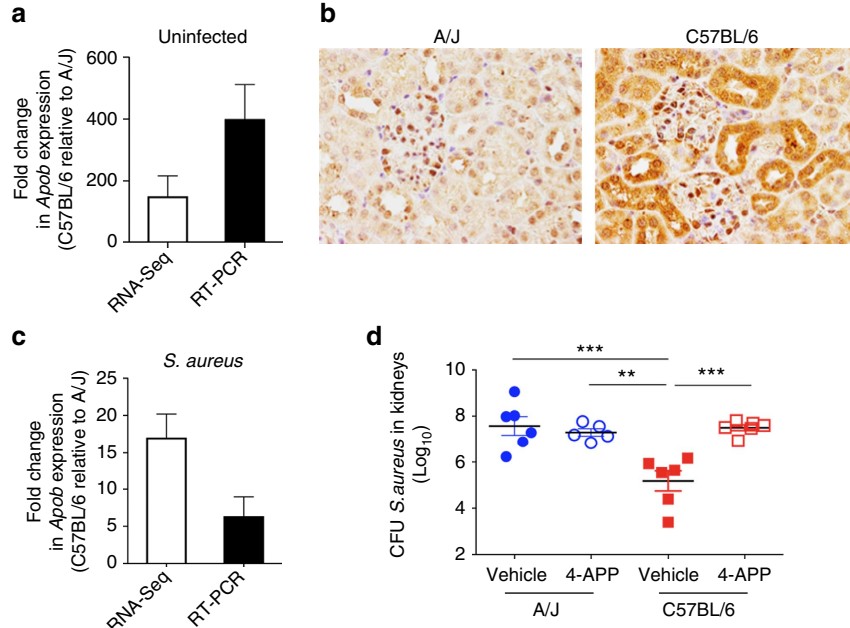

**Figure 5 | Apolipoprotein B (ApoB) contributes to resistance against *S. aureus* bloodstream infection in C57BL/6 mice. (a)** Relative fold change of *Apob* expression values in the kidneys of uninfected C57BL/6 respect to those in the kidneys of uninfected A/J mice determined by either RNA-Seq (white bars) or qRT–PCR (black bars). Each bar represents the mean relative fold change ± s.d. of triplicates. (**b**) Expression of ApoB in kidneys of A/J (left) and C57BL/6 (right) mice determined by immunostaining of kidney tissue using specific antibodies against ApoB. Magnification X40. (**c**) Relative fold change of *Apob* expression values in the kidneys of *S. aureus*-infected C57BL/6 in comparison to those in the kidneys of *S. aureus*-infected A/J mice at 48 h of infection determined by either RNA-Seq (white bars) or qRT–PCR (black bars). Each bar represents the mean relative fold change ± s.d. of triplicates. (**d**) Bacterial loads in the kidneys of A/J (blue symbols) and C57BL/6 (red symbols) mice treated with 4-Aminopyrazolo[3,4-d]pyrimidine (4-APP) (open symbols) or with vehicle alone (solid symbols) at 48 after intravenous inoculation with $2 \times 10^7$ CFU of *S. aureus* strain SH1000. Each symbol represents the bacterial counts determined in an individual mouse and the horizontal lines represent the average ± s.d. for each mouse strain ($n = 6$, $t$-test, **$P < 0.01$, ***$P < 0.001$).

**Table 3 | *S. aureus* gene expression during infection of 4-APP-treated and MyD88-deficient C57BL/6 mice.**

| Locus tag | Gene symbol | Description | Relative fold change in 4-APP-treated to vehicle-treated C57BL/6 mice (mean (s.e.)) | Relative fold change in MyD88-deficient to wild type C57BL/6 mice (mean (s.e.)) |
|---|---|---|---|---|
| SAOUHSC_02265 | *agrA* | Accessory gene regulator protein A | −1.60 (0.22) | −1.62 (0.5) |
| SAOUHSC_02566 | *sarR* | Accessory regulator R | −1.83 (0.22) | −1.23 (0.09) |
| SAOUHSC_01121 | *hla* | Alpha-hemolysin | −3.32 (1.56) | −18.06 (0.16) |
| SAOUHSC_02971 | *aur* | Zinc metalloproteinase aureolysin | −2.31 (0.96) | −2.59 (0.75) |
| SAOUHSC_00988 | *sspA* | Glutamyl endopeptidase | −2.19 (0.01) | −2.32 (0.46) |
| SAOUHSC_00561 | *vraX* | Protein VraX | −3.91 (2.76) | 1.11 (0.04) |
| SAOUHSC_00436 | *gltD* | Glutamate synthase subunit beta | −2.41 (1.11) | −6.03 (1.66) |
| SAOUHSC_01803 | *aapA* | D-serine/D-alanine/glycine transporter | 1.21 (2.32) | 1.69 (0.29) |
|  | RNAIII | Regulatory RNA | −1.80 (0.53) | −2.52 (0.08) |

Level of expression of a subset of genes by *S. aureus* during infection of 4-APP-treated compared with vehicle-treated C57BL/6, respectively MyD88-deficient compared with wild-type C57BL/6 mice determined by quantitative reverse transcription–PCR (qRT–PCR).

epithelial barrier[52]. The strong induction of genes encoding key exoproteases, including the metalloproteinase aureolysin (*aur*), serine proteases (*splA*, *sspA*, *splE* and *splF*), staphostatin B (*sspC*) and cysteine protease (*sspB*), further accentuates the importance of immune evasion for *S. aureus* survival in C57BL/6 mice. These proteases can cleave and degrade components of the complement system[53] and can inhibit neutrophil chemotaxis[54]. The expression of these virulence determinants is orchestrated through transcriptional and post-transcriptional regulation by regulatory systems. The expression of RNAIII, a major regulator of these factors[38], was higher expressed by *S. aureus* during infection of C57BL/6 mice than in A/J mice, while the expression of the autocatalytic sensory transduction system did not differ between

both mouse strains. This apparent discrepancy could, however, be explained by greater transcript abundance of SarR in *S. aureus* infecting C57BL/6 mice, which mitigates the expression of the *agr* P2 operon, while having no apparent effect on the expression of RNAIII (ref. 55). Moreover, SarR enhances the expression of genes encoding several extracellular proteases regulated by RNAIII (ref. 39) and, therefore, may act in synergy with RNAIII to boost the expression of virulence factors, required for survival under the strong immune pressure in C57BL/6 mice. The strong immune pressure in the resistant mice could also explain the greater expression of several genes of the cell wall stress stimulon by *S. aureus* infecting C57BL/6 mice. It has been shown that the magnitude of cell wall stimulon induction strongly

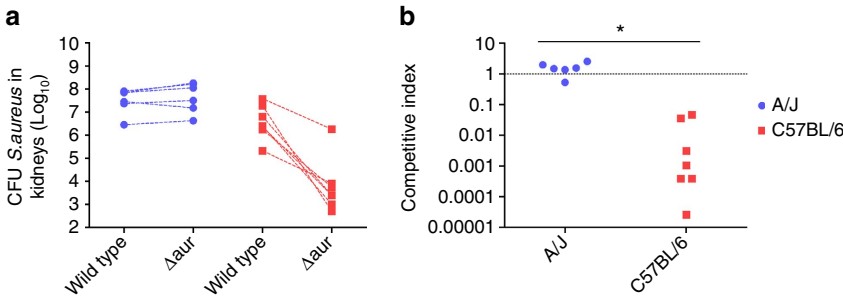

**Figure 6 | *In vivo* competitive fitness of aureolysin-deficient (*Δaur*) and wild type *S. aureus* during infection of C57BL/6 or A/J mice. (a)** C57BL/6 and A/J mice were injected intravenously with a 1:1 mixtures of *Δaur* and wild type *S. aureus* 8325-4 containing a total of approximately $4 \times 10^{7}$ bacteria. The number of bacterial cells in kidneys homogenates was determined at 48 h of infection. Symbols representing *Δaur* and wild type *S. aureus* 8325-4 bacteria from the same animal are connected by a broken line. (**b**) Competitive index (CI), representing the ratio of the recovered *Δaur* divided by the recovered wild type bacteria, within resistant C57BL/6 and susceptible A/J mice ($n = 7$, *t*-test,\**P* < 0.001).

depends on the concentration of cell wall damaging agents[56], which might indicate that *S. aureus* is exposed to a higher concentration of these damaging agents during infection of C57BL/6 than during infection of A/J mice.

To further corroborate the strong dependence of *S. aureus* expression of virulence factors on the strength of the host immune defense we investigated the effect of reducing the innate resistance of C57BL/6 mice to *S. aureus* either by chemotherapeutic reduction of ApoB concentration or by genetic deletion of the adaptor molecule MyD88 on the pathogen's gene expression. The results showed that reduction of resistance to infection in C57BL/6 mice resulted in a shift of *S. aureus* gene expression. The genes encoding the virulence factors alpha-hemolysin (*hla*), aureolysin (*aur*) and a glutamyl endopeptidase (*sspA*) as well as the virulence regulators SarR (*sarR*) and RNAIII were expressed to a lesser extent by *S. aureus* infecting the more susceptible 4-APP-treated or MyD88-deficient C57BL/6 mice than during infection of normal C57BL/6 mice. Together these results provide evidence of a relationship between host resistance and pathogen expression of virulence factors.

After having demonstrated the differential expression of virulence genes by *S. aureus* in response to different levels of host resistance during infection, we investigated the impact of this dependence on the efficacy of anti-virulence approaches. Targeting aureolysin, a gene that was expressed by *S. aureus* to a higher extent during infection of C57BL/6 than during infection of A/J mice, resulted in significant reduction of *S. aureus* fitness in C57BL/6 mice (*P* < 0.05, *t*-test), but did not affect the bacterial fitness during infection of A/J mice. These findings highlight that the efficacy of an anti-virulence strategy against *S. aureus* will depend on the level of expression of the targeted virulence factor. Given that we show effects of differential expression of virulence factors on the efficiency of targeted interference in a defined model using two inbred mouse strains, it can be assumed that the more diverse situation in humans would even further complicate the application of anti-virulence strategies.

In summary, by combining the transcriptional response of the pathogen and of the resistant and susceptible hosts in a single experimental system, we provided evidence of the impact of host intrinsic variability on the gene expression of *S. aureus* during infection. Differences in the capacity of C57BL/6 and A/J mice to control *S. aureus* resulted in different amounts of bacteria in the organs, which might affect bacterial gene expression via growth phase as well as quorum-sensing-dependent regulation and consequently result in the different expression of virulence factors. This information is essential when searching for novel anti-virulence targets. Although a number of anti-virulence

strategies targeting *S. aureus* toxins as well as the quorum-sensing regulator *agr* have been tested in pre-clinical mouse models with promising results[57–59], a common limitation of theses studies is the determination of treatment efficacy using a single host background. The findings of our study highlight the risk of drawing definitive conclusions for *in vivo* efficacy from such pre-clinical studies using a single host strain and emphasize the necessity to evaluate anti-virulence strategies in a range of host backgrounds. Furthermore, heterogeneity in the expression of virulence factors within the bacterial population infecting an individual host has been demonstrated in *S. aureus* during *in vivo* infection[60]. This will pose an additional problem for anti-virulence strategies, since only the specific population expressing the virulence factors targeted by the anti-virulence approach will be affected, leaving intact the rest of the bacterial population. This is an important issue that needs to be addressed in future studies.

## Methods

**Bacterial strains.** The *S. aureus* strains used in this study were strain SH1000 (ref. 61), the progenitor strain 8325-4 and the aureolysin-deficient (8325-4Δaur) *S. aureus* mutant strain[62]. Bacteria were grown to mid-log phase in brain heart infusion medium (BHI) at 37 °C with shaking (120 r.p.m.), collected by centrifugation, washed with sterile PBS and diluted to the required concentration for injection. The number of viable bacteria was determined after serial dilution and plating on blood agar.

**Mice and infection model.** Pathogen-free 10 weeks old female C57BL/6 and A/J mice were purchased from Harlan-Winkelmann (Envigo, The Netherlands) and had similar body weight (21 ± 1.9 g for A/J and 20.5 ± 0.8 g for C57BL/6 mice). Mice were infected intravenously with $4 \times 10^{7}$ colony forming units (CFU) of *S. aureus* strain SH1000 in 100 µl of PBS via a lateral tail vein. For determination of bacterial numbers in the kidneys, mice were killed by $CO_2$ asphyxiation at 48 h after bacterial inoculation, kidneys were removed and homogenized in PBS. Serial 10-fold dilutions of kidney homogenate were plated on blood agar plates. Bacterial colonies were counted after incubation at 37 °C for 24 h and calculated as CFU per kidneys.

For competition experiments, mice were intravenously inoculated with a 1:1 mixture of *S. aureus* strain 8325-4 and 8325-4Δaur mutant strain containing a total of ∼$4 \times 10^{7}$ bacteria. Mice were killed at 48 h of infection and the amount of each bacterial strain was determined by plating kidney homogenates in the absence (wild type + Δaur strain) or in the presence (Δaur strain) of 7.5 µg ml$^{-1}$ Erythromycin. The competitive index (CI) in the mixed infection was defined as the Δaur/wild type ratio in the infected kidneys at 48 h of infection. The experiment was repeated independently three times.

In some experiments, mice were treated intraperitoneally with 2.5 µg ml$^{-1}$ of 4-Aminopyrazolo[3,4-d]pyrimidine (4-APP, Sigma-Aldrich) in 0.9% NaCl solution or with vehicle alone (0.9% NaCl) at 48 and 24 h before bacterial inoculation. Individual mice were randomly assigned to the treatment or control group. Bacterial injections were performed blindly. Mice were then infected with $2 \times 10^{7}$ bacteria of *S. aureus* strain SH1000, killed at 48 h of infection and the amount of each bacterial strain was determined by plating on blood agar. The experiment was repeated independently three times.

Animal experiments were performed in strict accordance with the German regulations of the Society for Laboratory Animal Science (GV- SOLAS) and the European Health Law of the Federation of Laboratory Animal Science Associations (FELASA) and animals were excluded from further analysis if killing was necessary according to the human endpoints established by the ethical board. All experiments were approved by the ethical board Niedersächsisches Landesamt für Verbraucherschutz und Lebensmittelsicherheit, Oldenburg, Germany (LAVES; permit N. 33.9-42502-04-13/1195).

**Sample preparation and fixation for RNA-seq.** For RNA-seq experiments, three infected biological replicates from each mouse strain were analysed. Three uninfected, mock-treated replicates of each mouse strain were included as a reference to assess host gene expression changes induced by the infection. Each replicate consisted of pooled equimolar amounts of RNA extracted from the kidneys of five mice. To stabilize the RNA expression profiles, kidneys removed from infected and uninfected A/J and C57BL/6 mice were stored in 3 ml of RNAlater (Ambion) at 4 °C overnight.

**Total RNA extraction, purification and rRNA depletion.** Whole kidneys were mechanically disrupted using a Polytron disperser (Kinematica) in 2 ml of sterile PBS supplemented with 1% β-mercaptoethanol (Sigma-Aldrich) to denature RNases released on cell disruption and processed as previously described[63]. Samples were further treated using lysing matrices B (MP Biomedicals) to achieve efficient lysis of bacterial cells released from the homogenized tissue. Total RNA was purified from the disrupted samples using a classical phenol:chloroform:isoamyl alcohol method and concentrated by ethanol precipitation. Contaminating genomic DNA was removed by DNase treatment using 4 U Turbo DNase (Ambion) on 5 μg of total RNA for 30 min at 37 °C. To limit RNA degradation, 40 U of RiboGuard RNase Inhibitor (Epicentre) were added to the samples. RNA integrity was checked using an Agilent 2100 Bioanalyzer (Agilent Technologies). ERCC RNA Spike-In Mixes (Thermo Fisher Scientific) were added according to the manufacturer's guidelines to determine the dynamic range and lower limit of detection of the Illumina RNA-sequencing platform and control for experimentally derived errors in the generated datasets. DNA-depleted RNA samples were then depleted of murine and *S. aureus* ribosomal RNA using the RiboZero epidemiology kit (Epicenter) according to the manufacturer's protocol. Following ethanol precipitation at − 20 °C for 5 h, RNA quantity and rRNA depletion efficiency were accessed using the Agilent 2100 Bioanalyzer (Agilent Technologies) and NanoDrop 1000 spectrophotometer (Thermo Scientific).

**Illumina library preparation and sequencing.** The cDNA libraries were prepared using the ScriptSeq v2 RNA-seq library preparation kit (Epicentre) according to the manufacturer's instructions. Briefly, 50 ng of RNA were fragmented in fragmentation solution for 5 min at 85 °C and cDNA was synthesized using random hexamer primer including a 5′-end tagging sequence. RNA was removed and the 3′-tagged cDNA was synthesized via Terminal-Tagging Oligos. The generated di-tagged cDNA was purified using the MinElute PCR Purification kit according to the manufacturer's instructions. Subsequently, adaptor-tagged RNA-Seq libraries were generated by amplification of the di-tagged cDNA using primers that add indices for sample multiplexing and Illumina adaptor sequences. Following removal of excess PCR Primers via Exonuclease I treatment, the generated libraries were purified using the MinElute PCR Purification kit according to the manufacturer's instructions. The cDNA libraries were size-selected by agarose gel electrophoresis and fragments between 180 and 650 bp were enriched by gel extraction using the QIAquick gel extraction kit to exclude remaining contamination by other RNA species. Library quality and size-selection was assessed using the Agilent 2100 Bioanalyzer. The cDNA libraries were sequenced using single-end sequencing (50 bp) on the Illumina HiSeq 2500 platform utilizing the TruSeq S.R. cluster kit, v3-cBot-HS (Illumina). Two libraries were multiplexed per lane and sequenced to 58 cycles in one direction. Between 84 and 139 million reads were recorded per library.

**Data processing.** Raw sequenced reads were quality filtered and trimmed for Illumina-adapter contamination using fastq-mcf[64]. Remaining reads were aligned either to the genome of the *S. aureus* strain 8325-4, manually revised for the changes of strain SH1000 (ref. 21), or to the *Mus musculus* reference genome GRcm38.p3 using STAR[65]. Minimal sequence matches of 30 nucleotides with a minimal sequence homology of 93,33% were included. Mapped reads were collapsed using samtools[66] and counted using HTSeq[67]. Raw mapped read counts were normalized to transcripts per million (TPM).

**Quantitative RT–PCR.** Total RNA was extracted from tissue samples using the phenol:chloroform:isoamyl alcohol method. DNA was removed by DNase treatment utilizing the Turbo DNA-free kit (Invitrogen). RNA samples were reverse transcribed and amplified using a SensiFAST SYBR No-ROX Kit (Bioline) following the manufacturer's recommendations. The primers used for quantitative RT–PCR are provided in Supplementary Table 3. Thermal cycling conditions for *hld*/RNAIII, *sarR*, *hla*, *sspA*, *aur*, *vraX*, *gltB*, *aapA*, *apoB* and *β-actin* quantification

consisted of reverse transcription for 20 min at 45 °C, initial denaturation for 5 min at 95 °C, followed by 40 cycles of 20 s at 95 °C (denaturation), 20 s at 60 °C (annealing) and 20 s at 72 °C (elongation). Annealing temperature for the 16 s primers was 70 °C. Data were normalized against the housekeeping gene *16 s* for *S. aureus* and *β-actin* for *apoB*. Fold change values were calculated by the Pfaffl equation, in which the expression ratio is estimated by $(E_{target})^{\Delta Ct\ target\ (control\ -\ experimental)}/(E_{ref})^{\Delta Ct\ ref\ (control\ -\ experimental)}$. For *apoB*, the fold change between the expression of *apoB* in the kidneys of C57BL/6 was expressed relative to that in A/J mice. For *S. aureus* values were expressed as the fold change between the expression of a specific gene by *S. aureus* during the infection of 4-APP-treated C57BL/6 mice in respect to the expression by *S. aureus* during infection of vehicle-treated C57BL/6 mice.

**Apolipoprotein B immunohistochemistry.** Kidney samples were collected from uninfected C57BL/6 and A/J mice for immunohistochemistry. Tissues were fixed in 10% formalin and embedded in paraffin. Immunohistochemistry was performed using a polyclonal rabbit anti-mouse apolipoprotein B-specific antibody (Abcam ab20737). Paraffin sections were rehydrated through graded alcohols. For blocking of the endogenous peroxidase, formalin-fixed, paraffin-embedded tissue sections were treated with 0.5% $H_2O_2$ diluted in methanol for 30 min at room temperature (RT). Subsequently, sections were heated in 10 mM Na-citrate buffer pH 6.0 for 20 min in a microwave oven (800 W). Following incubation with 20% goat serum (obtained from University of Veterinary Medicine Hannover) for 30 min to block non-specific binding sites, sections were incubated with polyclonal rabbit anti-mouse apolipoprotein B-specific antibody (Abcam ab20737, dilution 1:50) for 1.5 h at RT. Rabbit serum (Sigma-Aldrich; R 4505) was used as negative control. Thereafter, sections were treated for 30 min at RT with the secondary goat anti-rabbit antibody (Vector Laboratories BA1000, dilution 1:200). Slides were subsequently incubated with the peroxidase-conjugated avidin-biotin complex (Vector Laboratories PK6100) for 30 min at RT. All antibodies were diluted in PBS. After visualization of the positive antigen-antibody reaction by incubation with 3.3-diaminobenzidine-tetrachloride (Fluka 32750) for 5 min, sections were counterstained with hematoxylin.

**Tissue ELISA.** Kidneys were taken from *S. aureus*-infected C57BL/6 and A/J mice at 48 h after bacterial inoculation or from mock-treated and homogenized in tissue lysis buffer containing 20 mM NaCl, 5 mM EDTA, 10 mM Tris, 1 mM PMSF, 1 μg/ml Leupeptin, 28 μg/ml Aprotinin at pH 7.4. Homogenized samples were centrifuged twice and the supernatant was collected and used for determination of interleukin 6 (IL-6), interleukin 1β (IL-1β), interleukin 10 (IL-10), Cxcl1, Cxcl2 and Ccl2 concentrations using matched antibody pairs and recombinant proteins as standards. Briefly, 96-well microtiter plates were coated overnight at 4 °C, with the corresponding purified anti-mouse capture monoclonal antibody (Biolegend 431305, 432602, 431412, R&D Systems DY452, DY453, BD Bioscience 555260) in the coating buffer ($Na_3PO_4$ pH 6.5, $Na_2CO_3$ pH 9.5 or PBS). Plates were washed and blocked with 1% bovine serum albumin-PBS or 10% fetal bovine serum-PBS. Tissue supernatants as well as standard were added in technical triplicates at previously established dilutions. Monoclonal anti-mouse detection antibody was added incubated for 1 h at 37 °C, washed and further incubated with avidin-horseradish peroxidase for detection (this step was omitted for Ccl2 quantification since the detection antibody was already HRP coupled). Plates were developed using TMB as substrate.

**Statistical analysis.** Statistical analyses were performed using PRIMER (Version 6.1.6, PRIMER-E; Plymouth Marine Laboratory), GraphPad Prism (Version 5.04, GraphPad Software, Inc.) and R (http://www.r-project.org). To assess the global similarity between biological replicates, a sample-similarity matrix was generated using euclidean distance measurement. The transcriptional profiles were compared using PCA and group-average agglomerative hierarchical clustering. PERMANOVA was used to determine the statistical significance of differences between the transcriptional profiles of the different groups. For the one-way PERMANOVA, the resemblance matrix was generated using euclidean distance measurement using type III (partial) sums of squares with a fixed effects sum to zero for mixed terms. Exact *P* values were generated using unrestricted permutation of the raw data. Pseudo-F statistic and generated *P* values were reported for each pair of conditions. Monte Carlo simulations were performed in the pairwise test function if low permutations were obtained. Differences between groups were considered significant if $P < 0.05$. Low read counts were filtered from the murine datasets applying a strict LODR-cutoff (limit of detection of ratio) as determined for a two-fold increase in transcript abundance by analysis of the ERCC spike-in mixes. To avoid misinterpretation resulting from the low sequencing depth of the *S. aureus* transcriptomes, a more stringent threshold was applied, where genes with transcripts levels below 0.05% (<16 copies per 33,228 reads) were filtered from the bacterial gene expression datasets. Due to the differences in sequencing depth different strategies were used for determination of differential gene expression analysis for host and pathogen. To identify genes differentially expressed by *S. aureus* between infection of A/J and C57BL/6 mice, the non-parametric NOISeq algorithm, whose sensitivity is less depended on the sequencing depth compared to other methods[68], was applied. A gene was considered differentially expressed by

*S. aureus* between infection of A/J and C57BL/6 mice when the FDR was < 0.05 (probability value > 0.95). The DESeq2 algorithm[22] was applied to identify genes differentially expressed between uninfected and *S. aureus*-infected A/J and C57BL/6 mice. A gene was considered significantly differential expressed between mice if the Benjamini–Hochberg adjusted *P* value was < 0.05 and the $\log_2$ fold change $\geq 1$. KEGG pathway enrichment analysis was performed using DAVID Informatics Resources[69]. KEGG pathways with a FDR $\leq 0.05$ were considered significantly enriched.

**Data availability.** The RNA-seq data that support the findings of this study have been deposited in the European Nucleotide Archive with the accession code PRJEB14649. All other relevant data are available from the corresponding author on request.

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

## Acknowledgements

We thank Tim Foster at the Trinity College, Dublin, for providing the 8325-4Δaur strain. We thank Dr Anne Fiebig at the Helmholtz Center for Infection Research, Braunschweig, for help with the bioinformatic analysis and Maria Voigt at the German Centre for Integrative Biodiversity Research (iDiv), Leipzig, for critical review of the manuscript. R.T. was partially supported by the President's Initiative and Network Fund of the Helmholtz Association of German Research Centres (HGF) under contract number VH-GS-202.

## Author contributions

R.T. and E.M. designed the study. R.T., O.G. and A.B. performed the experiments and analysis of the data. R.T. and E.M. wrote the manuscript.

## Additional information

**Competing financial interests:** The authors declare no competing financial interests.

