## [Peer Review File · Nature Communications]

Reviewers' comments:

Reviewer #1 (Remarks to the Author):

This is a very well-written, novel and interesting manuscript describing what appears to be the ability of *S. aureus* to modulate gene transcription, particularly of virulence factor genes, to 'match' the challenge presented by the host innate immune response. In general, the data are well-presented and the methodology appropriate for the questions addressed.

Suggestions for improvement are as follows:

1) Fig 7: Unlike Fig1b, in figure 7 there does not appear to be a significant difference in log CFU between C57BL/6 and A/J mice infected with wild-type *S. aureus* (same inoculum and same post-infection time point)? Please address this discrepancy.

2) The reported findings may have important implications regard the use of anti-virulence strategies as an approach to combatting antibiotic resistance. However, the relevance of targeting aureolysin in the 'blocking' studies is odd, although *aur* mutation when combined with *sae* mutation has been shown to decrease virulence in an osteomyelitis model (referenced in manuscript). Specifically, has aureolysin (or regulation of aureolysin) been suggested as a target for virulence inhibition?

3) ApoB-containing lipoprotein particles are produced in liver (apoB100) and gut (apoB48), so the relevance of measuring apoB message in the kidneys is unclear. How does message in the kidneys relate to serum apoB levels between C57BL/6 and A/J both pre- and post- infection. Quantitative measures of apoB protein (or at least non-HDL cholesterol) in serum are needed. These data may further strengthen the overall findings.

4) Tables 1,2: It would be helpful to include more traditional significance indicators (adjusted p value, FDR) in these tables or at a minimum explain how the Probability value relates to these other significance indicators.

5) Table 3: Please add a significance column to this table.

6) Figure 5a-c: Are these differences significant? If yes, please indicate on the graphs. If not, please clarify in the text that the differences were not statistically significant.

7) To eliminate body size as a potential contributor to infection response, please include the weights (or avg weight) of the C57BL/6 and A/J mice prior to infection.

8) Figure 1: Were CFUs also measured in blood, spleen or liver? Showing similar differences in CFUs in another organ/tissue (other than the kidney) would support the interpretation that this phenomenon is not tissue specific.

Reviewer #2 (Remarks to the Author):

In this manuscript the authors sought to determine how alterations in host immunity/resistance to infection affects the "quality and quantity" of pathogen gene expression in vivo. To this end, they infected A/J and C57BL6 mice with *S. aureus* as these strains of mice exhibit differentially susceptibility to this organism. RNAseq of both host and pathogen was performed revealing the host-pathogen combination profoundly affects the genes that change expression. The authors then identify one host factor (ApoB) and one bacterial factor (aureolysin) that are differentially expressed in these models and demonstrate that these factors are of variable importance

depending on the host-pathogen combination. From these results the authors conclude that host resistance influences the expression and importance of bacterial virulence factors which has implications for the design of anti-virulence strategies. It is a provocative and exciting model. However, the manuscript has a small number of major weaknesses which limit the ability to make conclusions and therefore limits the impact of the work.

1. The authors state that the different susceptibilities of the A/J and C57BL/6 mice can be assumed to be due to alterations in immune pressure between these animals. I do not agree with this assertion. The simple presence or absence of a receptor could be the explanation. This is a major concern as the contention that immune differences drive changes in gene expression is the basis for the entire experimental set-up. Comparing genetically identical mice where one mouse had specific arms of the immune system inactivated would be a stronger way to test their hypothesis. The authors moved in this direction with the apoB experiments, but much more could be done to enable comparisons across mice in genetically identical backgrounds.

2. Considering that infection of A/J and C57BL/6 mice resulted in differential bacterial burdens, how can the authors eliminate the impact of bacterial burden and phase of growth on gene expression? This is a well-established mechanism of global virulence gene regulation in *S. aureus*, and any transcriptional differences observed may be due to different bacterial burdens and not a factor of the presence or absence of specific immune components. This differential bacterial burden may also explain the "more severe systemic inflammation" observed in A/J mice.

3. Why were different bacterial inocula used for experiments in Figure 1 vs. those in Figure 6? Moreover, the authors should include 4-APP treatment of A/J mice to demonstrate that the effectiveness of this treatment is specific to a mouse that highly expresses ApoB. This type of reciprocal control experiment was performed when testing the impact of aureolysin (ie. both animals were infected with the mutant) which makes this absence particularly notable.

4. The Methods state that SH1000 and 8325-4 strains were used in this paper, however it is not clear which strains were used in which experiment. This needs to be stated explicitly. These two strains have sequence differences that affect the ability to compare across experiments.

Minor comments

1. Figure 1: The legend says 4E7 bacteria were inoculated but the figure says 5E7 bacteria were inoculated.

Reviewer #3 (Remarks to the Author):

This study by Thanert and colleagues reports the adoption of recently developed Dual RNA-seq to an in vivo model of *Staphylococcus aureus* infection. Simultaneous host-pathogen transcriptomic profiling enabled the authors to define infection-relevant differences in host gene expression between a resistant and a susceptible mouse strain. One finding is a strong pro-inflammatory immune response that was mounted primarily in the susceptible the host. Expression profiling of the bacterial pathogen identified a signature characteristic of *S. aureus* infection in the two host backgrounds. As a general trend, virulence genes showed higher expression during the infection of the susceptible host.

In the second part of the manuscript, the authors go on to demonstrate the relevance of their findings for the development of new antimicrobials. For instance, one of the host genes over-expressed in resistant compared to susceptible mice was ApoB. The authors show that treating susceptible mice with a drug inhibiting ApoB secretion indeed increased resistance against *S. aureus* infection. On the other hand, deleting the gene encoding aureolysin (a virulence factor over-expressed in *S. aureus* during the infection of resistant mice) led to a virulence defect in resistant but not in susceptible hosts. The authors therefore conclude that the success of anti-virulence strategies will strongly depend on the respective host background.

Overall, this is a very important study and a strong candidate for Nature Communications, pending successful revision along the lines below.

Major comments

- The introduction is unnecessarily lengthy, with plenty of repetitions and meandering text making it difficult for the reader to follow the rationale of the work. A major revision of the the introduction should also include a brief review of the current state of Dual RNA-seq as the entire study is strongly based on this method. The term "Dual RNA-seq" was introduced in 2012 already (PMID: 22890146) and Dual RNA-seq has since been successfully applied to diverse bacterial pathogens (examples include PMIDs: 24324615; 26578681; 25410299; 26789254), none of this is properly referenced at the moment. The authors should mention how their approach differs from those previous works, how library preparation/sequencing differed.
- Sample acquisition: RNA samples were taken from kidneys at 48 hrs after infection, with dramatically different bacterial loads in the kidneys of the two mouse strains (Fig. 1b). How can the authors distinguish between host expression changes that are mouse strain-specific in itself and those that simply arise from different bacterial counts (e.g. dose effects)? And if they cannot, they should at least mention that in the text and obviously be extremely careful when interpreting their findings.
- Data analysis: Ideally the exact same bioinformatics pipeline would be used to call differential gene expression for host and bacteria (i.e. the same tools, the same filters and cutoffs, etc.). The authors applied very different analysis pipelines (e.g. Bowtie 2 vs. STAR for alignment, DESeq2 vs. NOISeq for DGE calling, different cutoffs for sequence homology, etc.). As a general comment, data analysis should be homogenized for the two infection partners. If not possible, at least make clear in the main text that these differences exist and what might be the risks.
- Independent validation: The manuscript reports gene expression data as inferred from RNA-seq. All conclusions from the comparison between A/J and C57BL/6 infections are based on these values. However, there is hardly any independent confirmation of the RNA-seq-derived expression changes. Not a single bacterial transcript was measured by an independent method. On the host side, expression of only a single gene was independently determined (using qRT-PCR; Fig. 6) and the discrepancy between RNA-seq and qRT-PCR were dramatic (20-fold vs. 1,000-fold in Fig. 6b). At least a handful of (host and bacterial) genes should be independently measured (qRT-PCR, Northern blot), to evaluate how general this strong discrepancy between RNA-seq and gene-directed techniques really is (if so, there might be a problem with the RNA-seq data analysis). We note that there are qRT-PCR results for *S. aureus* transcripts in Table 3, but these come from a different experiment which was not analyzed by RNA-seq and consequently do not permit a direct comparison between fold-changes derived from the two methods. In addition, some validation on the protein level (at least for the central factors, e.g. ApoB) would be useful since an increase in mRNA abundance may not necessarily reflect elevated protein levels.
- Heterogeneity: The reported gene expression changes stem from bulk experiments. At least for a couple of bacterial genes, especially genes of the *arc* operon (Fig. 4) which are prominently discussed in the manuscript, heterogeneity in gene expression should be addressed. Could the authors use fluorescent reporter strains for infection of the two mouse strains and test whether the global induction in *arc* expression during A/J mouse infection as deduced from the bulk experiments holds true also on the single-cell level or, alternatively, if it is the result of a change in the ratio of *arc*-expressing vs. -non-expressing bacteria?

Minor comments

- In the Results section the authors focus on the differences between the two infections. For instance, they give numbers for those bacterial genes that show higher induction in one mouse

strain than the other. Likewise they give numbers for host genes higher expressed in one mouse strain than in the other. That is all fine but what the reader will also want to know is the overlap between the two infections (for host and bacteria).

- For competition experiments between Δ aur and wild-type Staph. only the Δ aur strain carries an antibiotics resistance marker. It appears that the resistance cassette itself had little or no impact on the fitness/growth of the bacterium, for example, both strains replicate to similar levels in A/J mice (Fig. 7a). It would, however, be cleaner if the resistance cassette was additionally swapped between the two strains without affecting the overall results (e.g. the fitness defect of Δ aur in C57BL/6 mice; Fig. 7a+b).

- Fig. 2a+b and entire Fig. 3 show plots of the RNA-seq screen and could be moved to the supplementary material.

- Fig. 2c is very confusing. Are these the pathways enriched in infected A/J mice compared to infected C57BL/6 mice? Or pathways generally enriched in infected compared to uninfected mice? Also the style of the diagram (bar charts and dots plotted on top) is difficult to interpret. Why not use heatmaps, on both the pathway level and the gene level?

- PCA stands for principal component analysis (not principal coordinates analysis; see page 21).

- There are many typos in the text. An (incomplete) list is given below.

- o Page 2: "... of the pathogens in vivo gene expression." should be corrected for "... of the pathogen's in vivo gene expression."

- o Page 4: "These characteristics make *S. aureus* one of the most dangerous and intractable infectious pathogen worldwide." should be corrected for: "These characteristics make *S. aureus* one of the most dangerous and intractable infectious pathogens worldwide."

- o Page 9: "...utilize arginine as a energy source..." should be corrected for: "...utilize arginine as an energy source..."

- o Page 14: "...virulence factors essential for the pathogens survival..." should be corrected for "...virulence factors essential for the pathogen's survival..."

- o Page 19: "Each replicated consisted of pooled kidneys..." should be corrected for "Each replicate consisted of pooled kidneys..."

“Response to Referees”

Reviewer #1

1) Fig 7: Unlike Fig1b, in figure 7 there does not appear to be a significant difference in log CFU between C57BL/6 and A/J mice infected with wild-type *S. aureus* (same inoculum and same post-infection time point)? Please address this discrepancy.

Answer: In the experiments depicted in Fig. 7, C57BL/6 and A/J mice were infected with 4×10^7 CFU of a mixture of the wild type and aureolysin-deficient *S. aureus* strain at a ratio of 1:1 and the CFU in this figure shows the amount of each bacterial strain recovered from the kidneys of infected mice at 48 h of infection. The total amount of bacteria in the kidneys resulting from the addition of CFU corresponding to the wild type strain plus the CFU corresponding to the aureolysin-deficient *S. aureus* mutant strain was $1.03 \times 10^7 \pm 5.4 \times 10^6$ for C57BL/6 mice and $1.3 \times 10^8 \pm 4.3 \times 10^7$ for AJ mice. The total number of *S. aureus* bacteria in the kidneys of C57BL/6 mice in the experiments depicted in Fig. 7 was significantly different from the total number of *S. aureus* in the kidneys of A/J mice ($p = 0.0138$) and comparable to the numbers shown in Fig. 1b. This information has been included in the revised manuscript (page 13).

2) The reported findings may have important implications regard the use of anti-virulence strategies as an approach to combatting antibiotic resistance. However, the relevance of targeting aureolysin in the 'blocking' studies is odd, although *aur* mutation when combined with *sae* mutation has been shown to decrease virulence in an osteomyelitis model (referenced in manuscript). Specifically, has aureolysin (or regulation of aureolysin) been suggested as a target for virulence inhibition?

Answer: Our intention was not to use aureolysin as a target to develop an anti-infective strategy against *S. aureus* but rather to evaluate whether the level of expression of a given virulence factor may have an impact on the efficacy of an anti-virulence strategy that targets this specific virulence determinant. We chose “aureolysin” in our study because it was differentially expressed by *S. aureus* during infection of C57BL/6 and during infection of A/J mice. In fact, any virulence factor that is expressed by *S. aureus* during infection may be a potential target for anti-virulence strategies since its expression indicates that it is required for bacterial survival within the host. However, the results of our study demonstrated that the expression of virulence factors by *S. aureus* during infection differs between hosts and, therefore, the effect of blocking these virulence factors as therapy will be of benefit only for those infected hosts where the bacterial virulence factor is expressed.

3) ApoB-containing lipoprotein particles are produced in liver (*apoB100*) and gut (*apoB48*), so the relevance of measuring *apoB* message in the kidneys is unclear. How does message in the kidneys relate to serum *apoB* levels between C57BL/6 and A/J both pre- and post- infection. Quantitative measures of *apoB* protein (or at least non-HDL cholesterol) in serum are needed. These data may further strengthen the overall findings.

Answer: ApoB is not only expressed in liver and gut but it was also shown to be expressed in

the kidneys, in particular by tubular epithelial cells (Krzystanek *et al.*, J Biol Chem. 2010, 285:10583-90). Because in our study the differences in bacterial loads between A/J and C57BL/6 mice were observed in the kidneys and not in blood, we have performed additional experiments to determine the expression of ApoB at the protein level in the kidneys of A/J and C57BL/6 by immunohistochemistry. The results demonstrate that the expression of ApoB at the protein level is much higher in the kidneys of C57BL/6 mice than in this organ of A/J mice, faithfully reflecting the mRNA data. A new panel has been added to Figure 5 (Fig. 5b) showing these results.

4) Tables 1,2: It would be helpful to include more traditional significance indicators (adjusted p value, FDR) in these tables or at a minimum explain how the Probability value relates to these other significance indicators.

Answer: We have added a column to Table 1, Table 2 and Supplementary Table 6 showing the FDR values.

5) Table 3: Please add a significance column to this table.

Answer: In this table, we show the mean relative fold-change of the level of expression of 11 genes expressed by *S. aureus* during infection of 4-APP-treated C57BL/6 mice respect to the level of expression of these genes in *S. aureus* during infection of untreated mice, as calculated by the Pfaffl equation. As we analyze the relative changes in genes expression of one biological condition to a reference/baseline we provided the standard error of the mean (SEM) as a common indication for the variance of the fold changes between independent experiments or the precision of our data. We have also added a new column to this table showing the fold-change in the level of expression of the 11 genes in *S. aureus* during infection of MyD88-deficient C57BL/6 mice versus the level of expression of these genes by *S. aureus* during infection of wild type C57BL/6 mice, since this new experimental data was requested by the Reviewer #2.

6) Figure 5a-c: Are these differences significant? If yes, please indicate on the graphs. If not, please clarify in the text that the differences were not statistically significant.

Answer: The significance has been added to the figures as suggested by the reviewer.

7) To eliminate body size as a potential contributor to infection response, please include the weights (or avg weight) of the C57BL/6 and A/J mice prior to infection.

Answer: The A/J and C57BL/6 mice used in all experiments included in the manuscript had similar body weight (21 ± 1.9 g for A/J and 20.5 ± 0.8 g for C57BL/6 mice). This information has been included in the Materials and Methods section of the revised manuscript (page 18).

8) Figure 1: Were CFUs also measured in blood, spleen or liver? Showing similar differences in CFUs in another organ/tissue (other than the kidney) would support the interpretation that this phenomenon is not tissue specific.

Answer: The bacterial loads have been also determined in the liver of A/J and C57BL/6 mice and the results show that, similarly to the kidneys, the amount of *S. aureus* in the liver of A/J mice was significantly greater than in the liver of C57BL/6 mice. A new graph showing this

data has been included in Figure 1 (Fig. 1c) of the revised manuscript.

Reviewer #2

1. The authors state that the different susceptibilities of the A/J and C57BL/6 mice can be assumed to be due to alterations in immune pressure between these animals. I do not agree with this assertion. The simple presence or absence of a receptor could be the explanation. This is a major concern as the contention that immune differences drive changes in gene expression is the basis for the entire experimental set-up. Comparing genetically identical mice where one mouse had specific arms of the immune system inactivated would be a stronger way to test their hypothesis. The authors moved in this direction with the apoB experiments, but much more could be done to enable comparisons across mice in genetically identical backgrounds.

Answer: We designed our experimental set-up using two different mouse strains that strongly differ in their level of susceptibility to *S. aureus* infection to reflect the human situation, where variation in the susceptibility to infection in the general population is a complex trait, defined by several genetically determined and acquired characteristics of the immune system. We think that the influence of the complex nature of host susceptibility on the transcriptional response of *S. aureus* is best reflected by using the genetically diverse background of A/J and C57BL/6, where it is known that several differences in the make-up of the immune system are contributing to the differential susceptibility to *S. aureus*.

Nevertheless, to address the reviewers concerns, we have performed additional experiments comparing the levels of gene expression by *S. aureus* during infection of genetically identical mice where one strain had specific arms of the immune system inactivated. We have chosen wild type C57BL/6 mice versus MyD88-deficient C57BL/6 mice for this purpose. MyD88 is a cytoplasmic adaptor molecule essential for the signaling of IL-1R/TLR family. C57BL/6 mice deficient in the expression of MyD88 exhibit significantly greater levels of susceptibility to *S. aureus* than wild type C57BL/6 mice (Takeuchi *et al.*, J Immunol. 2000,165:5392-6). The results of this experiment show that the expression level of *hld/RNAlII*, *sarR*, *hla*, *sspA*, *aur*, *vraX*, *gltB* and *aapA* is strongly different between *S. aureus* infecting Myd88-deficient mice and *S. aureus* infecting wild type C57BL/6 mice. These results clearly demonstrated that reducing resistance to infection by interfering with TLR signaling has a profound effect on the transcriptional response of *S. aureus* in an otherwise genetically identical background. These results have been included in Table 3 and in the Results section of the revised manuscript (page 12).

2. Considering that infection of A/J and C57BL/6 mice resulted in differential bacterial burdens, how can the authors eliminate the impact of bacterial burden and phase of growth on gene expression? This is a well-established mechanism of global virulence gene regulation in *S. aureus*, and any transcriptional differences observed may be due to different bacterial burdens and not a factor of the presence or absence of specific immune components. This differential bacterial burden may also explain the "more severe systemic inflammation" observed in A/J mice.

Answer: Our aim in this study was to evaluate how differences in the levels of host resistance to *S. aureus* in general influence the bacterial expression of virulence factors during infection. Our intention was not to point out specific immune components since resistance/susceptibility to infection are complex traits and may be influenced by many

different host factors.

We don't eliminate an impact of bacterial burden and phase of growth on *S. aureus* gene expression. Indeed, the incapacity of the immune response to control *S. aureus* in a susceptible host such as is the case of A/J mice in our study, can result in progressive bacterial multiplication and increased levels systemic inflammation. In our opinion this is reflective of the situation in humans, where differences in the capacity of the immune system to control invading pathogens varies significantly between individuals. These differences result in mild infections in some patients, while others develop severe infections associated with higher bacterial burdens when confronted with the same pathogen. Consequently, the combination of progressive multiplying bacteria and a mounting but permissive inflammatory response at the site of infection will create an environment that can strongly influence the bacterial gene expression. We don't exclude the possibility that these environments impact the gene expression of *S. aureus* via density or growth phase dependent regulation. However, we highlight that the different environments at the site of infection in A/J and C57BL/6 mice have a particularly strong influence on the expression of virulence factors and thereby could greatly affect the effectiveness of anti-virulence strategies.

3. Why were different bacterial inocula used for experiments in Figure 1 vs. those in Figure 6? Moreover, the authors should include 4-APP treatment of A/J mice to demonstrate that the effectiveness of this treatment is specific to a mouse that highly expresses ApoB. This type of reciprocal control experiment was performed when testing the impact of aureolysin (ie. both animals were infected with the mutant) which makes this absence particularly notable.

Answer: In preliminary experiments we observed that 4-APP treatment increased the susceptibility of C57BL/6 mice to *S. aureus* infection and, according to the humane endpoints established in our ethical permit, we had to euthanize the 4-APP-treated mice before 48 h after inoculation with a dose of 4×10^7 CFU. To reduce the animal suffering and distress and to follow the regulations of the German Society for Laboratory Animal Science (GV-SOLAS) and the European Health Law of the Federation of Laboratory Animal Science Associations (FELASA), we have slightly reduced the dose of the bacterial inoculum to 2×10^7 CFU in the experiments presented in Fig. 6. This inoculum reduced the distress of 4-APP-treated mice and maintained the differences in bacterial burdens between untreated and 4-APP-treated mice at 48 h of infection.

The experiment involving 4-APP treatment of A/J mice requested by the reviewer has been performed and the results show that 4-APP-treated A/J mice are as susceptible as those treated with the vehicle alone. Thus, inhibition of ApoB does not change the level of susceptibility of A/J mice. This information has been included in the revised manuscript in Fig. 5d.

4. The Methods state that SH1000 and 8325-4 strains were used in this paper, however it is not clear which strains were used in which experiment. This needs to be stated explicitly. These two strains have sequence differences that affect the ability to compare across experiments.

Answer: The specific *S. aureus* strains used in the different experiments have been indicated in the revised manuscript.

Minor comments

1. Figure 1: The legend says 4E7 bacteria were inoculated but the figure says 5E7 bacteria were inoculated.

Answer: This has been corrected in the revised manuscript.

Reviewer #3

Major comments

The introduction is unnecessarily lengthy, with plenty of repetitions and meandering text making it difficult for the reader to follow the rationale of the work. A major revision of the introduction should also include a brief review of the current state of Dual RNA-seq as the entire study is strongly based on this method. The term "Dual RNA-seq" was introduced in 2012 already (PMID: 22890146) and Dual RNA-seq has since been successfully applied to diverse bacterial pathogens (examples include PMIDs: 24324615; 26578681; 25410299; 26789254), none of this is properly referenced at the moment. The authors should mention how their approach differs from those previous works, how library preparation/sequencing differed.

Answer: The introduction has been shortened and a brief review of the history and current state of Dual RNA-seq has been added to the introduction of the revised manuscript as suggested. A much more detailed description of sample processing, library preparation, sequencing strategy and *in silico* analysis has been included in the Materials and Methods section (page 19-21) to facilitate a better comparison to published work.

• *Sample acquisition: RNA samples were taken from kidneys at 48 hrs after infection, with dramatically different bacterial loads in the kidneys of the two mouse strains (Fig. 1b). How can the authors distinguish between host expression changes that are mouse strain-specific in itself and those that simply arise from different bacterial counts (e.g. dose effects)? And if they cannot, they should at least mention that in the text and obviously be extremely careful when interpreting their findings.*

Answer: We will always have the problem of the differences on bacterial loads when comparing hosts that strongly vary in their levels of resistance to a certain pathogen. We know from previous experiments that the difference in bacterial loads between resistant C57BL/6 and susceptible A/J mice was also observed when different doses of inoculum were used, since the level of resistance is strongly connected to the capacity of the host immune defense to control the invading pathogen. This is, however, also the situation in humans, where the capacity of the immune system to control invading pathogens at the site of entry strongly varies among individuals, resulting in some individuals developing only mild self-limiting infection, while others experience severe invasive infections in response to the same pathogen. Our intention in this manuscript was to investigate how different levels of host resistance to infection, whether genetically determined in C57BL/6 versus A/J mice, resulting from prior interventions (Apolipoprotein B depletion) or introduced by genetic deletion of a specific receptor in Myd88-deficient C57BL/6 (new data in the revised manuscript), can influence the transcriptional response of *S. aureus* during infection. As already pointed out in response to reviewer #2 (Comment 2) we don't eliminate an impact of the different bacterial

burdens between C57BL/6 and A/J mice on gene expression of host and pathogen. Indeed, the differences in the capacity of C57BL/6 and A/J mice to control *S. aureus* results in different amount of bacteria in the organs and different levels of inflammation. We demonstrated that these completely different environments at the site of infection in A/J and C57BL/6 mice have a particularly strong influence on the transcriptional response of *S. aureus* and, therefore, could greatly affect the expression of virulence factors and the effectiveness of anti-virulence strategies. A sentence clarifying this issue has been included in the Discussion section of the revised manuscript (page 16/17).

• Data analysis: Ideally the exact same bioinformatics pipeline would be used to call differential gene expression for host and bacteria (i.e. the same tools, the same filters and cutoffs, etc.). The authors applied very different analysis pipelines (e.g. Bowtie 2 vs. STAR for alignment, DESeq2 vs. NOISeq for DGE calling, different cutoffs for sequence homology, etc.). As a general comment, data analysis should be homogenized for the two infection partners. If not possible, at least make clear in the main text that these differences exist and what might be the risks.

Answer: We have re-analyzed the data and changed the bioinformatic tool for aligning the RNA-seq data to the *S. aureus* genome using now STAR for both host and pathogen and homogenized the filtering procedure (93.33% sequence homology) for host and bacterium. This entailed small changes (between 0.17% and 2.19%) in the number of uniquely mapped reads for the different biological replicates. Consequently, Figures 3 and 4, Table 1 and 2 as well as the Supplementary Tables 5 and 6 have changed marginally. We checked that changes in the number of uniquely mapped sequences are not due to reads mapping cross-species that are now picked up by the softening of the cutoff (100 % to 93.33 % sequence homology), since the reads newly aligned to the *S. aureus* genome did not map to the *M. musculus* reference.

While we see the need to homogenize the data analysis in a dual RNA-seq experiment as far as possible, the specific implications of the individual characteristics of both datasets, specifically for the statistical analysis, have to be considered. The sequencing depths of the different biological replicates of *S. aureus* varied to significantly greater extend in comparison to that of the host (maximal fold difference in the number of uniquely mapped reads: *S. aureus* 161.97 fold; *M. musculus* 2.66 fold). Compared to other algorithms for differential gene expression calling, the non-parametric NOISeq algorithm is less dependent on the sequencing depth and dispersion in the analyzed datasets (DESeq, edgeR, baySeq) (Tarazona *et al.*, Genome research 2011, 21.12 2213-2223; Sonesson and Delorenzi, BMC bioinformatics 2013, 14(1), 1), while producing reliable results (Zheng and Etsuko, BMC bioinformatics 2013, 14.Suppl 13: S7; Nookaew *et al.*, Nucleic Acids Res. 2012, gks804). Because of the high variation between replicates in sequencing depth, we chose to analyze the bacterial datasets using NOISeq. However, such variation is not present in the host datasets, therefore, a parametric algorithm with higher statistical power like DESeq2 was more appropriate to analyze these data. A sentence pointing out the remaining difference of the DGE calling algorithm has been introduced in the revised manuscript (Methods page 24).

• Independent validation: The manuscript reports gene expression data as inferred from RNA-seq. All conclusions from the comparison between A/J and C57BL/6 infections are based on these values. However, there is hardly any independent confirmation of the RNA-seq-derived expression changes. Not a single bacterial transcript was measured by an independent method. On the host side, expression of

only a single gene was independently determined (using qRT-PCR; Fig. 6) and the discrepancy between RNA-seq and qRT-PCR were dramatic (20-fold vs. 1,000-fold in Fig. 6b). At least a handful of (host and bacterial) genes should be independently measured (qRT-PCR, Northern blot), to evaluate how general this strong discrepancy between RNA-seq and gene-directed techniques really is (if so, there might be a problem with the RNA-seq data analysis). We note that there are qRT-PCR results for *S. aureus* transcripts in Table 3, but these come from a different experiment which was not analyzed by RNA-seq and consequently do not permit a direct comparison between fold-changes derived from the two methods.

Answer: Independent validation of gene expression by RT-PCR has been performed for a set of bacterial and host genes. For the investigated factors, the results of the gene-directed method closely reflect the results of the RNA-seq analysis. A new figure (Supplementary Figure 3) has been included in the Supplementary material section of the revised manuscript.

In addition, some validation on the protein level (at least for the central factors, e.g. ApoB) would be useful since an increase in mRNA abundance may not necessarily reflect elevated protein levels.

Answer: Validation of a set of differentially expressed genes at the protein levels has been performed, including the expression of ApoB in the kidneys of A/J and C57BL/6 mice (Figure 5b). A new figure (Supplementary Figure 4) showing these results has been included in the Supplementary material section of the revised manuscript.

• Heterogeneity: The reported gene expression changes stem from bulk experiments. At least for a couple of bacterial genes, especially genes of the *arc* operon (Fig. 4) which are prominently discussed in the manuscript, heterogeneity in gene expression should be addressed. Could the authors use fluorescent reporter strains for infection of the two mouse strains and test whether the global induction in *arc* expression during A/J mouse infection as deduced from the bulk experiments holds true also on the single-cell level or, alternatively, if it is the result of a change in the ratio of *arc*-expressing vs. -non-expressing bacteria?

Answer: Heterogeneity in gene expression within the bacterial population has been already reported in *S. aureus* during *in vivo* infection (Liese *et al.*, Cell Microbiol 2013, 15(6):891-909). In that report, the authors investigated the spatial regulation of *S. aureus agr* and *sar* gene expression *in vivo* on a single cell level using an Agr-reporter strains and intravital two-photon microscopy. They found high heterogeneity in the expression of these regulatory systems within the bacterial population infecting the skin. Heterogeneity in the expression of virulence factors within the bacterial population will pose an additional problem for anti-virulence strategies since only the specific population expressing the virulence factors targeted by the anti-virulence approach will be affected. This is an extremely important issue that we will address in future studies. A sentence addressing this issue has been added to the Discussion section of the revised manuscript (page 17).

Minor comments

• In the Results section the authors focus on the differences between the two

infections. For instance, they give numbers for those bacterial genes that show higher induction in one mouse strain than the other. Likewise they give numbers for host genes higher expressed in one mouse strain than in the other. That is all fine but what the reader will also want to know is the overlap between the two infections (for host and bacteria).

Answer: Venn diagrams showing the numbers of unique and overlapping genes in the bacteria and host transcriptome analysis have been included in Figure 2 and Figure 3 of the revised manuscript. The genes induced to a similar extent by *S. aureus* in both A/J and C57BL/6 mice are those included in Supplementary Table 6 with no significant difference in gene expression ($p \geq 0.05$).

• For competition experiments between Δ aur and wild-type *Staph.* only the Δ aur strain carries an antibiotics resistance marker. It appears that the resistance cassette itself had little or no impact on the fitness/growth of the bacterium, for example, both strains replicate to similar levels in A/J mice (Fig. 7a). It would, however, be cleaner if the resistance cassette was additionally swapped between the two strains without affecting the overall results (e.g. the fitness defect of Δ aur in C57BL/6 mice; Fig. 7a+b).

Answer: As the referee already rightfully points out, the results indicate that the cassette itself is having no influence on the *in vivo* fitness, since we would otherwise expect a similar fitness defect in the A/J mice. Nevertheless, we have compared the *in vitro* growth of the Δ aur mutant and wild type strain and found no growth defect of the Δ aur mutant. Moreover, this particular Δ aur *S. aureus* mutant strain has been extensively used by other groups together with the corresponding wild type strain in *in vivo* experiments (Shaw *et al.*, Microbiology 2004, Jan 1;150(1):217-28; Calander *et al.*, Microbes and Infection 2004, 6.2: 202-206).

• Fig. 2a+b and entire Fig. 3 show plots of the RNA-seq screen and could be moved to the supplementary material.

Answer: Fig. 2a,b and Fig.3 have been moved to the Supplementary Material in the revised manuscript (Supplementary Figure 1 and Supplementary Figure 2).

• Fig. 2c is very confusing. Are these the pathways enriched in infected A/J mice compared to infected C57BL/6 mice? Or pathways generally enriched in infected compared to uninfected mice? Also the style of the diagram (bar charts and dots plotted on top) is difficult to interpret. Why not use heatmaps, on both the pathway level and the gene level?

Answer: Following the reviewers suggestions, Figure 2c has been exchanged for a new figure comprising a heatmap depicting the mean fold change of gene expression and a bar chart showing the corresponding number of genes significantly higher expressed for each KEGG pathway of the category 'immune system' in the kidneys of infected compared to uninfected A/J and C57BL/6 mice.

• PCA stands for principal component analysis (not principal coordinates analysis; see page 21).

Answer: This has been corrected in the revised manuscript.

• There are many typos in the text. An (incomplete) list is given below.

o **Page 2:** *"... of the pathogens in vivo gene expression." should be corrected for "... of the pathogen's in vivo gene expression."*

Answer: This has been corrected in the revised manuscript.

o **Page 4:** *"These characteristics make S. aureus one of the most dangerous and intractable infectious pathogen worldwide." should be corrected for: "These characteristics make S. aureus one of the most dangerous and intractable infectious pathogens worldwide."*

Answer: This has been corrected in the revised manuscript.

o **Page 9:** *"...utilize arginine as a energy source..." should be corrected for: "...utilize arginine as an energy source..."*

Answer: This has been corrected in the revised manuscript.

o **Page 14:** *"...virulence factors essential for the pathogens survival..." should be corrected for "...virulence factors essential for the pathogen's survival..."*

Answer: This has been corrected in the revised manuscript.

o **Page 19:** *"Each replicated consisted of pooled kidneys..." should be corrected for "Each replicate consisted of pooled kidneys..."*

Answer: This has been corrected in the revised manuscript.

REVIEWERS' COMMENTS:

Reviewer #1 (Remarks to the Author):

The authors have addressed this reviewer's initial concerns. There are two corrections that should be made to the revised manuscript:

1) Pg 13, there is no Figure 6 (lines 345 and 349). Presumably, Fig 6 is labeled as Fig 7. This should be corrected.

2) In the response to reviewers (Rev 1 #6) the authors say that significance has been added to Figs 5a-c. However, on page 12 lines 300-302 of the revised manuscript, the authors state that gene encoding apoB was "expressed to a significantly greater extent in the kidneys of C57BL/6 than in the kidneys of A/J mice at both gene expression (Fig. 5a) and protein level (Fig. 5b), before and after infection (Fig. 5c)." Since significance is not shown on 5a or 5c, and 5b is a microscopy image, the word 'significant' should be removed.

Reviewer #2 (Remarks to the Author):

The authors have addressed most of my prior comments which has resulted in an improved paper. However, I still have one concern with a statement made in the text.

In response to the prior review, the authors acknowledge that bacterial growth phase may be the driving force behind changes in gene expression, but they argue that this closely mimics the situation in humans. That argument seems reasonable, yet they still state in the manuscript that "These findings indicate that *S. aureus* needs to express a greater repertoire of virulence factors to evade the host defense mechanisms during infection of C57BL/6 than in A/J mice." I do not believe this is supported by the data in the paper and the authors should soften this conclusion and acknowledge the likelihood that growth phase is the major driving factor behind these changes in gene expression.

Reviewer #3 (Remarks to the Author):

This study by Medina and coworkers has much improved since we last saw it. Particularly, we appreciate the author's efforts to provide additional data to support their previous RNA-seq data. We recommend some remaining minor issues be addressed by amendments to the text prior to publication.

- Is it really justified to say that "... *S. aureus* ENCOUNTERS completely different physiological conditions during infection of A/J and C57BL/6 mice..." (p. 8, line 200)? In fact, it looks (see Suppl. Fig. S1b, cluster #1) as if prior to infection, the host transcriptomes of the two mouse strains are very similar. Rather it seems as if *S. aureus* infection makes these hosts provide different environments (see Suppl. Fig. S1b, clusters #2+3). Since the authors looked only at a single time point after infection (2 days) and do not follow infection over time, it is impossible to distinguish cause from consequence. That is, from the current dataset alone it cannot be deduced whether differences in *S. aureus* expression cause the dramatically different host responses at 2 days p.i. or whether the different host responses make *S. aureus* express its genes differently. Therefore the authors should consider rephrasing some of their sentences in the manuscript (e.g. p. 8, line 200; p. 8, line 207).

- We appreciate the independent validation experiments to support the RNA-seq data and note

that the overall correlation between RNA-seq and qRT-PCR was very good (Suppl. Fig. S3). Still we think that the authors should give an explanation for why the expression of apoB, the central host gene in their study, was an exception as the fold-changes deduced from the two methods differed markedly (Fig. 5c).

- The way the manuscript is currently written, the experiment in MyD88 knockout mice comes a bit out of the blue (p. 12, line 318), making the reader wonder how MyD88 deletion is related to ApoB inhibition. The two experiments should be put in better context, e.g. by mentioning that a highly similar set of *S. aureus* genes was de-regulated in both experiments; currently the text only says that deletion of MyD88 "... had a strong influence on the *S. aureus* virulence gene expression..." (p. 8, lines 323-324).

- Figure 4c: It is unusual to combine primary data with a model scheme in a single panel. The bar charts in panel c having a font size different from those in panels a and b look strange. Why not split panel c into two separate panels – one with the bar charts and one with the regulatory model – and homogenize the font sizes throughout the entire figure 4?

- There are still numerous typos and mislabels throughout the text. Examples include:

- o p. 3, line 69: "it is believe" must be "it is believed"

- o p. 4, line 99: "is strongly influence" must be "is strongly influenced"

- o p. 5, line 120: "the host-dependence bacterial expression" must be "the host-dependent bacterial expression"

- o p. 6, line 140: "variability in the host response to infection affect the transcriptional response" must be "variability in the host response to infection affects the transcriptional response"

- o p. 6, line 148: "Fig. 1c" must be "Fig. 1d"

- o p. 17, line 450: "infecting a individual host" must be "infecting an individual host"

- o p. 19, line 506: "from each mouse strains" must be "from each mouse strain"

- o p. 19, line 507: A full stop is missing at the end of this sentence.

- o p. 19, line 512: "... and mRNA enrichment" should be changed to "... and rRNA depletion"

- o p. 20, line 522: "using a Agilent 2100 Bioanalyzer" must be "using an Agilent 2100 Bioanalyzer"

- o p.20, line 535: "fragmented fragmentation solution" must be "fragmented in fragmentation solution"

- o The legend to new figure 6 is still labelled with "figure 7".

“Response to Referees”

Reviewer #1

1) Pg 13, there is no Figure 6 (lines 345 and 349). Presumably, Fig 6 is labeled as Fig 7. This should be corrected.

Answer: The Figure number has been corrected in the Figure legend.

2) In the response to reviewers (Rev 1 #6) the authors say that significance has been added to Figs 5a-c. However, on page 12 lines 300-302 of the revised manuscript, the authors state that gene encoding apoB was “expressed to a significantly greater extent in the kidneys of C57BL/6 than in the kidneys of A/J mice at both gene expression (Fig. 5a) and protein level (Fig. 5b), before and after infection (Fig. 5c).” Since significance is not shown on 5a or 5c, and 5b is a microscopy image, the word ‘significant’ should be removed.

Answer: The word “significant” has been removed from the text as suggested.

Reviewer #2

In response to the prior review, the authors acknowledge that bacterial growth phase may be the driving force behind changes in gene expression, but they argue that this closely mimics the situation in humans. That argument seems reasonable, yet they still state in the manuscript that “These findings indicate that S. aureus needs to express a greater repertoire of virulence factors to evade the host defense mechanisms during infection of C57BL/6 than in A/J mice.” I do not believe this is supported by the data in the paper and the authors should soften this conclusion and acknowledge the likelihood that growth phase is the major driving factor behind these changes in gene expression.

Answer: As requested by the reviewer, we have modified the conclusion on page 10:

“These findings indicate that *S. aureus* expressed greater levels of virulence factors during infection of C57BL/6 than during infection of A/J mice, which is most probably driven by the different growth phase of the bacteria in the two mouse strains.”

Reviewer #3

Is it really justified to say that “... S. aureus ENCOUNTERS completely different

physiological conditions during infection of A/J and C57BL/6 mice...” (p. 8, line 200)? In fact, it looks (see Suppl. Fig. S1b, cluster #1) as if prior to infection, the host transcriptomes of the two mouse strains are very similar. Rather it seems as if S. aureus infection makes these hosts provide different environments (see Suppl. Fig. S1b, clusters #2+3). Since the authors looked only at a single time point after infection (2 days) and do not follow infection over time, it is impossible to distinguish cause from consequence. That is, from the current dataset alone it cannot be deduced whether differences in S. aureus expression cause the dramatically different host responses at 2 days p.i. or whether the different host responses make S. aureus express its genes differently. Therefore the authors should consider rephrasing some of their sentences in the manuscript (e.g. p. 8, line 200; p.8, line 207).

Answer: The sentences have been rephrased in the manuscript as suggested by the reviewer:

Page 8, line 200: “Taken together, these findings suggest that the micro-environment in the infected tissue is highly different between A/J and C57BL/6 mice, which could significantly affect the expression of virulence determinants by *S. aureus*.”

Page 8, line 207: “In parallel, we analyzed the transcriptome of *S. aureus* during the infection of resistant C57BL/6 and susceptible A/J mice in order to determine the impact of the different physiological conditions present at the site of infection on the pathogen’s transcriptional response.”

We appreciate the independent validation experiments to support the RNA-seq data and note that the overall correlation between RNA-seq and qRT-PCR was very good (Suppl. Fig. S3). Still we think that the authors should give an explanation for why the expression of apoB, the central host gene in their study, was an exception as the fold-changes deduced from the two methods differed markedly (Fig. 5c).

Answer: We have optimized the RT-PCR conditions and repeated the RT-PCR reactions for detection of *ApoB* expression in samples from infected and uninfected mice. While the detected fold change for the uninfected mice was highly similar to our earlier results, the calculated fold change for the infected mice did vary significantly from our previous analysis. This discrepancy originated from a technical problem in the amplification of the reference/housekeeping gene in the samples from the infected mice in the previous RT-PCR analysis, which went unnoticed. The new results of the fold change in *apoB* expression from the infected mice are more comparable to those obtained by RNA-seq. The graphs in Fig. 5a and Fig. 5c have been edited now showing the new results. We apologize for the mistake and thank the reviewer for insisting on this point.

The way the manuscript is currently written, the experiment in MyD88 knockout mice comes a bit out of the blue (p. 12, line 318), making the reader wonder how MyD88 deletion is related to ApoB inhibition. The two experiments should be put in better context, e.g. by mentioning that a highly similar set of S. aureus genes was de-regulated in both experiments; currently the text only says that deletion of MyD88 “... had a strong influence on the S. aureus virulence gene expression...” (p. 8, lines 323-324).

Answer: The paragraph has been modified in the manuscript.

“These genes were also found to be differentially expressed between *S. aureus* infecting C57BL/6 mice and *S. aureus* infecting C57BL/6 mice deficient in the expression of MyD88, an adaptor molecule that is essential for the signaling of IL-1R/TLR family (Table 3). Since MyD88-deficient mice are more susceptible to *S. aureus* than wild type C57BL/6 mice⁴⁵, these observations further demonstrated the remarkable influence of the level of host resistance on the transcriptional response of *S. aureus* during.”

Figure 4c: It is unusual to combine primary data with a model scheme in a single panel. The bar charts in panel c having a font size different from those in panels a and b look strange. Why not split panel c into two separate panels – one with the bar charts and one with the regulatory model – and homogenize the font sizes throughout the entire figure 4?

Answer: We would like to keep the bar charts showing the levels of gene expression (bar charts) within the scheme. This makes it easier for the readers to understand the complexity of the data. Nevertheless, we have homogenized the font size throughout the entire Figure 4 as suggested by the reviewer.

There are still numerous typos and mislabels throughout the text. Examples include:

p. 3, line 69: “it is believe” must be “it is believed”

Answer: Corrected

p. 4, line 99: “is strongly influence” must be “is strongly influenced”

Answer: Corrected

p. 5, line 120: “the host-dependence bacterial expression” must be “the host-dependent bacterial expression”

Answer: Corrected

p. 6, line 140: “variability in the host response to infection affect the transcriptional response” must be “variability in the host response to infection affects the transcriptional response”

Answer: Corrected

p. 6, line 148: “Fig. 1c” must be “Fig. 1d”

Answer: Corrected

p. 17, line 450: “infecting a individual host” must be “infecting an individual host”

Answer: Corrected

p. 19, line 506: “from each mouse strains” must be “from each mouse strain”

Answer: Corrected

p. 19, line 507: A full stop is missing at the end of this sentence.

Answer: Corrected

p. 19, line 512: “... and mRNA enrichment” should be changed to “... and rRNA depletion”

Answer: Corrected

p. 20, line 522: “using a Agilent 2100 Bioanalyzer” must be “using an Agilent 2100 Bioanalyzer”

Answer: Corrected

p.20, line 535: “fragmented fragmentation solution” must be “fragmented in fragmentation solution”

Answer: Corrected

The legend to new figure 6 is still labelled with “figure 7”.

Answer: Corrected